# UPL: Uncertainty-aware Pseudo-labeling for Imbalanced Transductive Node Classification

## Abstract

Graph-structured datasets often suffer from class imbalance, which complicates node classification tasks. In this work, we address this issue by first providing an upper bound on population risk for imbalanced transductive node classification. We then propose a simple and novel algorithm, Uncertainty-aware Pseudo-labeling (UPL). Our approach leverages pseudo-labels assigned to unlabeled nodes to mitigate the adverse effects of imbalance on classification accuracy. Furthermore, the UPL algorithm enhances the accuracy of pseudo-labeling by reducing training noise of pseudo-labels through a novel uncertainty-aware approach. We comprehensively evaluate the UPL algorithm across various benchmark datasets, demonstrating its highly competitive performance compared to existing state-of-the-art methods.

## 1 Introduction

Graphs are fundamental data structures that are universally utilized in various applications. The flexibility and expressiveness of graph representations enable us to model complex relationships and interactions within several types of data across numerous fields. From social networks, where individuals are connected by various kinds of relationships, to biological networks that represent protein interactions, graphs provide a powerful way to encapsulate the inherent complexities of many natural and human-made systems (Mohammadrezaei et al., 2018; Ying et al., 2018; Perozzi et al., 2016; Hamilton et al., 2017). In tackling the task of learning from graph data, Graph Neural Networks (GNNs) play an important role (Hamilton et al., 2017; Kipf & Welling, 2016a; Velickovic et al., 2017). GNNs leverage the rich relational information within graphs, allowing for the effective propagation of information through the network structure and the ability to generalize deep learning approaches to non-Euclidean data. However, despite their advantages, GNNs encounter significant challenges when dealing with imbalanced data in transductive scenarios where the unlabeled nodes are utilized during the training phase, a common scenario in real-world graphs (Park et al., 2022). In many graph-based applications, some classes of nodes are vastly underrepresented compared to others, leading to a problem in the learning process (Song et al., 2022; Park et al., 2022). In particular, this imbalance can result in biased models towards the majority class, which can also depend on graph structure. For instance, in social networks, minority groups might be underrepresented in friendship suggestions; rare protein interactions could be overlooked in biological networks.

The problem of imbalance in graphs thus poses unique challenges that require specialized strategies to address. Standard imbalance handling techniques, such as re-weighting and re-sampling, may work in fields such as image classification but face significant obstacles when applied to graphs. Re-weighting methods, which assign higher weights to minority nodes, do not alter the connectivity patterns of these nodes, leaving their neighborhood influences unchanged. Re-sampling strategies, intended to balance class representation, can disrupt the graph's structure, potentially altering critical message-passing pathways and affecting the semantic integrity of node classes and embedded features.

Despite these developments, balancing the enhancement of minority class representation against the integrity of majority classes remains a technical challenge due to overemphasis on minority classes which can degrade the model's ability to accurately represent more common classes (Yan et al., 2024; Song et al., 2022; Park

et al., 2022; Jervakani et al., 2024). Furthermore, the generalization error and population risk study of imbalanced node classification is overlooked. Our contributions in this work can be summarized as follows:

- **Theoretical Generalization Bound:** We derive an upper bound on population risk for imbalanced transductive node classification.

- **Uncertainty-aware Pseudo-Labeling (UPL):** We propose UPL, a novel algorithm that utilizes topology-aware uncertainty (Selective Edge Removal) and dual-threshold filtering to augment minority classes.

- **State-of-the-Art Performance:** Extensive experiments show UPL outperforms baselines on homophilic and heterophilic graphs while being faster than leading competitors like UNREAL.

## 2 Related works

**Imbalanced Graph Classification:** Due to the non-Euclidean characteristics of graph data, the imbalance problem in this type of data necessitates additional attention and specialized focus. Following the categorization of imbalance classification methodologies, we similarly categorize imbalanced graph classification models into two main groups: data-level and algorithm-level methods.

*Data-level methodologies* try to rebalance the learning environment by manipulating training data in feature or label spaces. Fundamental approaches such as over-sampling and under-sampling, prevalent in traditional class-imbalanced learning, require adaptation to suit the complexities inherent in graph data, characterized by intricate node dependencies and interconnections. Data-level methods could be subcategorized into data interpolation, adversarial generation, and pseudo-labeling (Ma et al., 2023). Data interpolation techniques, exemplified by SMOTE (Chawla et al., 2002) and its graph-oriented variant, GraphSMOTE (Zhao et al., 2021b), generate synthetic minority nodes by interpolating between existing node representations. Methods like GATSMOTE (Liu et al., 2022b) and GNN-CL (Li et al., 2024) enhance this process through attention mechanisms, ensuring quality edge predictions between synthetic and real nodes. Moreover, Mixup strategies, as seen in GraphMixup (Wu et al., 2022) and GraphENS (Park et al., 2022), introduce sophisticated blending techniques to combat neighbor memorization issues and prevent out-of-domain sample generation. Building upon these blending concepts, GraphSHA (Li et al., 2023a) addresses the "squeezed minority problem" by synthesizing harder minority samples. It utilizes a SemiMixup module to carefully enlarge minority decision boundaries in the latent space without deteriorating the topological subspaces of neighboring major classes.

*Algorithm-level methodologies* focus on refining learning algorithms to accommodate class imbalance effectively. Model refinement strategies adapt graph representation learning architectures to improve performance by incorporating class-specific adjustments and modules tailored to handle imbalance. For instance, approaches like ACS-GNN (Ma et al., 2022) and EGCN (Wang et al., 2022) modify aggregation operations within graph neural networks to prioritize minority class information, ultimately enhancing classification accuracy. Additionally, loss function engineering tries to design customized loss functions to prioritize minority class errors or encourage wider class separations, a challenging task given the connectivity properties of nodes in graphs. Recent innovations like ReNode (Chen et al., 2021) and TAM (Song et al., 2022) integrate graph topology information into loss function designs, showcasing advancements in addressing class imbalance within the graph context. Jervakani et al. (2024) proposed a KL regularization for imbalance node classification. Diverging from traditional class-rebalancing paradigms entirely, the BAT framework (Liu et al., 2023) approaches the problem through a topological augmentation lens. By leveraging model-based prediction uncertainty, BAT identifies and rectifies nodes critically influenced by ambivalent and distant message-passing, mitigating class-imbalance bias directly through dynamic structural adjustments.

**Generalization Error and Node Classification:** There are two different scenarios in node classification, including inductive and transductive scenarios. Inductive learning node classification involves (semi-)supervised learning on graph data samples where the test data are unseen during training. For Inductive node classification tasks, (Verma & Zhang, 2019) discussed the generalization error under node classification for GNNs via algorithm stability analysis in an inductive learning setting based on the SGD optimization algorithm. The work by Zhou & Wang (2021) extended the results in (Verma & Zhang, 2019) and found that increasing

the depth of Graph Convolutional Neural Network (GCN) enlarges its generalization error for the node classification task. In Transductive Node Classification, based on transductive Rademacher complexity, a high-probability upper bound on the generalization error of the transductive node classification task was proposed by Esser et al. (2021). The transductive modeling of node classification was studied in (Oono & Suzuki, 2020) and (Tang & Liu, 2023). (Cong et al., 2021) presented an upper bound on the generalization error of GCNs for node classification via transductive uniform stability, building on the work of (El-Yaniv & Pechyony, 2006). In contrast, our research focuses on the task of imbalanced transductive node classification and we provide an upper bound on population risk.

**Uncertainty in Self-training:** The integration of uncertainty estimation in GNNs has been extensively explored, particularly in self-training frameworks Wang et al. (2021); Yang et al. (2021); Zhao et al. (2021a). Early contributions, such as Li et al. (2018), introduced a simple strategy that selects the top K most confident nodes based on softmax probabilities for pseudo-labeling, expanding the labeled set iteratively. This approach was later refined by Sun et al. (2020), which employed a multi-stage self-training method, iteratively updating node labels to enhance GNN performance in sparse label settings. Extending these ideas, Yang et al. (2021) proposed a self-enhanced GNN framework that utilizes an ensemble of models to ensure consistency in pseudo-label predictions, thereby improving robustness against label noise. A common technique for uncertainty estimation in these methods is the use of entropy, as demonstrated in Cai et al. (2017); Zhang et al. (2021b), which measures prediction confidence to guide node selection. Building on this, Zhao et al. (2021a) introduced an entropy-aware self-training framework, incorporating an entropy-aggregation layer to account for graph structural information in uncertainty estimation. While traditional approaches in entropy-based uncertainty estimation, prioritize high-confidence nodes for pseudo-labeling, this practice might unintentionally reinforce distributional biases, as highlighted by Liu et al. (2022a), where over-reliance on highly certain nodes risks a distribution shift between the original labeled set and the expanded pseudo-labeled set. Furthermore, Li et al. (2023b) observed that nodes with very high confidence often contribute redundant information, limiting the diversity of the labeled dataset and potentially biased learning process. To address this concerning issue, in this work, we suggest to balance node selection across a broader confidence range to improve the performance of our model.

## 3   Preliminaries

**Notations:** We adopt the following convention for random variables and their distributions in the sequel. A random variable is denoted by an upper-case letter (e.g., $Z$), its space of possible values is denoted with the corresponding calligraphic letter (e.g., $\mathcal{Z}$), and an arbitrary value of this variable is denoted with the lower-case letter (e.g., $z$). We denote the set of integers from 1 to $N$ by $[N] \triangleq \{1, \ldots, N\}$; the set of measures over a space $\mathcal{X}$ with finite variance is denoted $\mathcal{P}(\mathcal{X})$. For a matrix $\mathbf{X} \in \mathbb{R}^{k \times q}$, we denote the $i-$th row of the matrix by $\mathbf{X}[i, :]$. The Euclidean norm of a vector $X \in \mathbb{R}^q$ is $\|X\|_2 := (\sum_{j=1}^{q} x_j^2)^{1/2}$. For a matrix $\mathbf{Y} \in \mathbb{R}^{k \times q}$, we let $\|\mathbf{Y}\|_\infty := \max_{1 \le j \le k} \sum_{i=1}^{q} |\mathbf{Y}[j, i]|$ and $\|\mathbf{Y}\|_F := \sqrt{\sum_{j=1}^{k} \sum_{i=1}^{q} \mathbf{Y}^2[j, i]}$.

**Information Measures:** The KL divergence $\mathrm{KL}(P\|Q)$ is given by $\mathrm{KL}(P\|Q) := \int_{\mathcal{Z}} \log\left(\frac{dP}{dQ}\right) dP$. We also define the cross-entropy between $P$ and $Q$, as $H(P, Q) = -\int_{\mathcal{Z}} \log(dP) dQ$.

**Graph data samples:** We consider transductive node classification for undirected graphs with $N$ nodes that have no self-loops or multiple edges. Inputs to GNNs are node samples, which are comprised of their features and graph adjacency matrices. We denote the space of node feature for a node in the graph by $\mathcal{X}$. We consider the adjacency matrix $\mathbf{A} \in \mathcal{A} \subset \{0, 1\}^{N \times N}$ to be fixed with maximum node degree for $i$-th class $\deg_i^{\max}$ and minimum node degree for $i$-th class $\deg_i^{\min}$. The input pair of $i$-th node feature with dimension $k$ is denoted by $\mathbf{x}_i$. The GNN output (label) is denoted by $y \in \mathcal{Y}$ where $\mathcal{Y}$ is the space of labels and $|\mathcal{Y}| = k$. Define $\mathcal{Z} = \mathcal{X} \times \mathcal{Y}$. Let $\mathbf{Z}_m = \{Z_i\}_{i=1}^{m} \in \mathcal{Z}$ denote the training of labeled dataset, where the $i-$th node sample is $Z_i = (X_i, Y_i)$. For transductive learning in node classification, we denote $\mathbf{Z}_u = \{Z_i\}_{i=1+m}^{m+u} \in \mathcal{Z}$ as the set of unlabeled dataset. We assume that the features of nodes are i.i.d samples from the the feature space $\mathcal{X}$. We also denote the matrix of node features in the graph with $\mathbf{X} \in \mathbb{R}^{N \times k}$. We denote the learned distribution over classes as $P(\hat{\mathbf{Y}}|\mathbf{X}) := \{P(\hat{Y} = j|\mathbf{X})\}_{j=1}^{k}$ where $P(\hat{Y} = j|\mathbf{X}) := \frac{1}{n} \sum_{i=1}^{n} P(\hat{Y} = j|X_i)$ is the

prediction of model for $j$-th class for given node-features $\{X_i\}_{i=1}^n$. Furthermore, we define $\mathbf{P}_k := \{P_j\}_{j=1}^k$ as the target class distribution where $P_j = \frac{|m_j|}{\sum_{i=1}^k |m_i|}$ shows the probability of $j$-th class within the dataset.

**Graph filters:** Graph filters are linear functions of the adjacency matrix, $\mathbf{A}$, defined by $\mathrm{G}_f : \mathcal{A} \mapsto \mathbb{R}^{N \times N}$ where $N$ is the size of the graph, see in (Defferrard et al., 2016). Graph filters model the aggregation of node features in a GNN. For example, the symmetric normalized graph filter proposed by Kipf & Welling (2016b) is $\mathrm{G}_f(\mathbf{A}) = \tilde{L} := \tilde{D}^{-1/2} \tilde{\mathbf{A}} \tilde{D}^{-1/2}$ where $\tilde{\mathbf{A}} = I + \mathbf{A}$ is the augmented adjacency matrix, $\tilde{D}$ is the degree-diagonal matrix of $\tilde{\mathbf{A}}$, and $I$ is the identity matrix. Furthermore, $\deg_i^{\max}$ and $\deg_i^{\min}$ are the maximum and minimum degree of nodes among $i$-th class of the graph with the augmented adjacency matrix, $\tilde{\mathbf{A}}$. Another normalized filter, a.k.a. random walk graph filter (Xu et al., 2019), is $\mathrm{G}_f(\mathbf{A}) = D^{-1}\mathbf{A} + I$, where $D$ is the degree-diagonal matrix of $\mathbf{A}$. The mean-aggregator is also a well-known aggregator defined as $\mathrm{G}_f(\mathbf{A}) = \tilde{D}^{-1}(\mathbf{A} + I)$. The sum-aggregator graph filter is defined by $\mathrm{G}_f(\mathbf{A}) = (\mathbf{A} + I)$.

**Loss function:** With $\mathcal{Y}$ the label space, the loss function $\ell : \mathbb{R} \times \mathcal{Y} \to \mathbb{R}$ is denoted as $\ell(\hat{y}, y)$, where $\hat{y}$ is defined as the output of model, e.g., $\ell(h_\theta(\mathrm{G}_f(\mathbf{A})[i, :]\mathbf{X}), y_i)$ is the loss function for $i$-th node in the graph where $h_\theta : \mathcal{X} \mapsto \mathcal{Y}$ is the hypothesis of our parameterized model which belongs to $\mathcal{H}$ with $\theta \in \Theta$.

**True and empirical risks:** The empirical risk function for labeled node samples $\mathbf{X}_m$ is given by

$$\mathrm{R}(\mathbf{Z}_m, h_\theta) := \frac{1}{m} \sum_{i=1}^m \ell(h_\theta(\mathrm{G}_f(\mathbf{A})[i, :]\mathbf{X}_m), y_i). \tag{1}$$

We also define the true risk as the average of loss function over unlabeled nodes, $\mathbf{X}_u$, in transductive learning,

$$\mathrm{R}(\mathbf{Z}_u, h_\theta) := \frac{1}{u} \sum_{i=m+1}^{m+u} \ell(h_\theta(\mathrm{G}_f(\mathbf{A})[i, :]\mathbf{X}_u), y_i). \tag{2}$$

We would like to study the performance of the model trained with the labeled data set $\mathbf{Z}_m = \{(\mathbf{X}_i, Y_i)\}_{i=1}^m$ and evaluated against the unlabeled dataset $\mathbf{Z}_u = \{(X_i, Y_i)\}_{i=m+1}^{m+u}$ loss, using transductive Rademacher complexity analysis (El-Yaniv & Pechyony, 2009).

## 4 The Generalization Error of Imbalanced Transductive Node Classification

This section provides a generalization error upper bound under an imbalance scenario for transductive node classification via Rademacher complexity analysis. All proof details are deferred to Appendix C.

For theoretical analysis, we need the following assumptions.

**Assumption 4.1** (Bounded node features). For every node, the features are contained in an $\ell_2$-ball of radius $B_f$. In particular, $\|\mathbf{X}[i;]\|_2 \leq B_f$ for all nodes.

**Assumption 4.2** (Bounded norm of parameters). There exists $U_F(i) \in \mathbb{R}$ such that $\|\theta_i\|_F \leq U_F(i)$ where $\theta_i$ is parameter matrix of $i$-th layer.

**Assumption 4.3** (Activation functions). The activation function $\varphi : \mathbb{R} \mapsto \mathbb{R}$ is 1-Lipschitz [1], so that $|\varphi(x_1) - \varphi(x_2)| \leq |x_1 - x_2|$ for all $x_1, x_2 \in \mathbb{R}$, and zero-centered, i.e., $\varphi(0) = 0$.

In theoretical analysis, we consider binary classification, based on (Bartlett et al., 2017), where $\gamma$-margin loss function for binary classification is proposed. For a positive $\gamma \in \mathbb{R}^+$, we have, $\ell_\gamma(y_1, y_2) = 0$ and $\ell_\gamma(y_1, y_2) = \min(1, 1 - \frac{y_1 y_2}{\gamma})$ for $y_1 y_2 \geq \gamma$ and $y_1 y_2 < \gamma$, respectively.

For transductive learning analysis, inspired by El-Yaniv & Pechyony (2009), we define empirical transductive Rademacher complexity as for $p \in [0, 1/2]$ and let the set $\boldsymbol{\epsilon} := \{\epsilon_i\}_{i=1}^{m+u}$ be i.i.d. with probability $P(\epsilon_i = 1) = P(\epsilon_i = -1) = p$ and $P(\epsilon_i = 0) = 1 - 2p$ for all $i \in [m + u]$ and assuming $p = \frac{m+u}{(m+u)^2}$,

$$\mathfrak{R}_{m+u}(\mathcal{H}) := \left(\frac{1}{m} + \frac{1}{u}\right) \mathbb{E}_{\boldsymbol{\epsilon}} \left[ \sum_{i=1}^{m+u} \epsilon_i h_\theta(x_i) \right].$$

---

[1] For the Tanh and Relu activation function satisfy this assumption.

Similarly, we can define the empirical transductive Rademacher complexity for $i$-th class with $\mathfrak{R}_{m_i+u_i}(\mathcal{H})$. Furthermore, in transductive learning, we consider $m_i$ and $u_i$ as labeled nodes and unlabeled nodes belong to the $i$-th class. We first provide an upper bound on population risk for transductive learning under the imbalance scenario.

**Proposition 4.4.** *Let $Q_i = (\frac{1}{u_i} + \frac{1}{m_i})$, $S_i = \frac{m_i+u_i}{(m_i+u_i-1/2)(1-1/(2\max(m_i,u_i)))}$ for $i = 1, 2$. Then, with probability at least $(1 - \delta/2)$ over the choice of the training set from nodes of graphs, for all $h_\theta \in \mathcal{H}$,*

$$R(\mathbf{Z}_u, h_\theta) \leq \sum_{i=1}^{2} R_\gamma(\mathbf{Z}_{m_i}, h_\theta) + \frac{u_i}{u}\Big(\frac{1}{\gamma}\mathfrak{R}_{m_i+u_i}(\mathcal{H}) + c_0 Q_i \sqrt{\min(m_i, u_i)} + \sqrt{\frac{S_i Q_i}{2}\log(1/\delta)}\Big),$$

*where $\mathfrak{R}_{m_i+u_i}(\mathcal{H})$ is the transductive Rademacher complexity for $i$-th class, $R_\gamma(\mathbf{Z}_{m_i}, h_\theta)$ is the empirical risk based on $\gamma$-margin loss for $i$-th class and $c_0 = \sqrt{\frac{32\ln(4e)}{3}}$.*

Next, we provide an upper bound on the transductive Rademacher complexity for general deep GNN.

**Proposition 4.5.** *Given Assumptions 4.1, 4.2 and 4.3, then the following upper bound holds on the transductive Rademacher complexity of node classification for GNN with depth $d$ and graph filter $\mathrm{G}_f(\mathbf{A})$,*

$$\mathfrak{R}_{m_i+u_i}(\mathcal{H}) \leq \Big(C_1\|\mathrm{G}_f(\mathbf{A})[i]\|_\infty^{2(d-1)}\Pi_{j=1}^d U_F(j)\Big)\sqrt{\frac{1}{m_k} + \frac{1}{u_k}},$$

*where $C_1 = B_f(\sqrt{2\log(2)d} + 1)$ and $\|\mathrm{G}_f(\mathbf{A})[i]\|_\infty$ is the infinite norm of graph filter among $i$-th class.*

Combining Proposition 4.5 with Proposition 4.4, we can derive the following upper bound on population risk.

**Theorem 4.6.** *Under the same Assumptions in Proposition 4.4 and Proposition 4.5, the following upper bound holds on population risk for imbalanced transductive node classification under a GCN model,*

$$R(\mathbf{Z}_u, h_\theta) \leq \sum_{i=1}^{2} R_\gamma(\mathbf{Z}_{m_i}, h_\theta) + c_0 Q_i \sqrt{\min(m_i, u_i)} + \sqrt{\frac{S_i Q_i}{2}\log(1/\delta)}\Big)$$

$$+ \frac{u_i}{u}\Big(\frac{B_f(\sqrt{2\log(2)d} + 1)\Pi_{j=1}^d U_F(j)}{\gamma}\sqrt{\frac{1}{m_i} + \frac{1}{u_i}} \times \Big(\frac{\deg_i^{\max} + 1}{\deg^{\min} + 1}\Big)^{d-1},$$

*where $U_F(i)$ is the maximum of Frobenius norm of $i$-th layer parameter.*

**Convergence rate discussion:** For large values of $u_i$ and $m_i$ ($i \in 1, 2$), the value of $S_i$ approaches one. Moreover, we find $Q_i = O(1/\min(m_i, u_i))$ and $\mathfrak{R}_{m_i+u_i}(\mathcal{H}) = O(\sqrt{\frac{1}{m_i} + \frac{1}{u_i}})$. Consequently, assuming $u_i \ll m_i$ for both $i = 1, 2$, yields an overall convergence rate of $O(1/u)$. Conversely, when $m_i \ll u_i$, the convergence rate becomes $O(\max(\frac{1}{\sqrt{m_1}}, \frac{1}{\sqrt{m_2}}))$. It's noteworthy that in the balanced case, where $u_1 = u_2$ and $m_1 = m_2$, assuming $m_i \ll u_i$ for $i = 1, 2$ recover the convergence rate applicable to balanced node classification, dependent on the maximum degree per class and the minimum degree.

**Maximum and Minimum Degrees:** In balance transductive node classification, e.g., (Oono & Suzuki, 2019; Cong et al., 2021; Tang & Liu, 2023), the upper bound on population risk is dependent on the maximum and minimum degree of all nodes. Our bound in Theorem 4.6 reveals that for an imbalance scenario, the upper bound depends on the maximum degree of nodes per class and the minimum degree of all graph nodes. Note that, this result can also be applied to balance scenario by assuming $m_i = m$ for all classes.

**Discussion:** Theorem 4.6 reveals that the population risk upper bound is fundamentally dominated by the number of samples in the minority class. This theoretical insight suggests a straightforward yet powerful approach: increasing the representation of minority classes can improve classification performance. While this observation emerges directly from our theoretical analysis, its implications are significant as it applies broadly to any transductive imbalanced node classification task. In the following section, we propose an algorithm that leverages this insight by strategically assigning pseudo-labels to unlabeled nodes, thereby effectively augmenting the minority class representation and improving node-classification performance. An extension of current results for multi-class is provided in App. K.

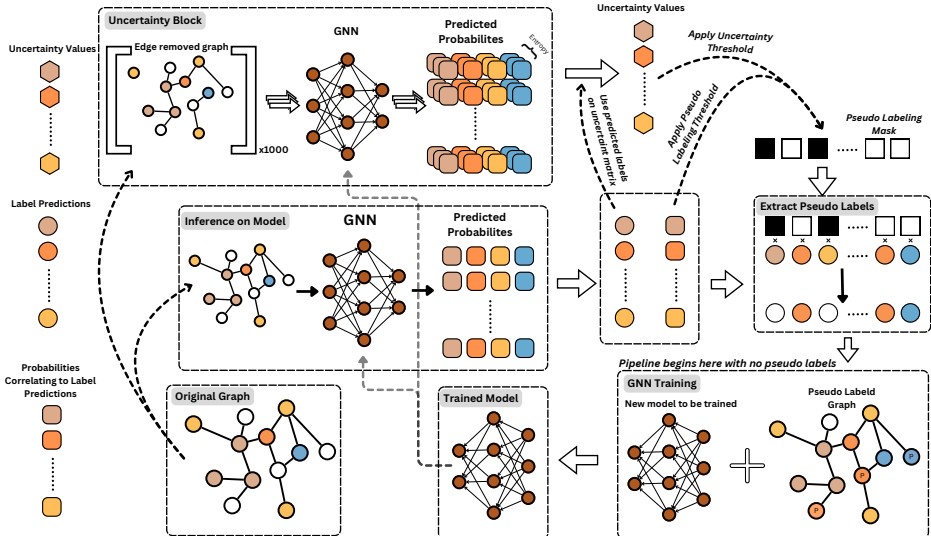

Figure 1: Pipeline of the UPL Algorithm

## 5 UPL Algorithm

In this section, we introduce the Uncertainty-aware Pseudo-labeling (UPL) Algorithm to mitigate the imbalance effect in transductive node classification.

In the UPL algorithm, we add pseudo-labels to unlabeled nodes of minority classes to mitigate the effect of imbalance node classification. Our UPL algorithm contains two blocks: uncertainty-aware pseudo-labeling, and balanced-softmax loss function. The pipeline of the UPL algorithm is shown in Figure 1. Our UPL algorithm involves an iterative training approach, leveraging the pseudo-labeling technique to enhance the model's performance on graph-based data.

### 5.1 Uncertainty-aware Pseudo-labeling

Pseudo-labeling is a semi-supervised learning technique that enhances model performance by generating synthetic labels for unlabeled data using a previously trained model. This method effectively expands the training dataset, allowing the model to learn from a broader range of examples and improve accuracy. Different strategies for Pseudo-labeling are proposed. In this work inspired by the technique in (Rizve et al., 2021), we propose an iterative uncertainty-aware pseudo-labeling method where the uncertainty is computed based on the graph's topology.

**Pseudo-labeling:** To refine the selection of pseudo-labels, we incorporate two thresholds: a lower threshold ($\eta_l$) and an upper threshold ($\eta_u$). As proposed by Rizve et al. (2021), the lower threshold selects high-confidence samples. To avoid overfitting to these high-confidence samples, we introduce an upper threshold, $\eta_u$, which selects samples with intermediate confidence levels. This can be seen as a band-pass filter on the nodes, including those within a specific probability range. This method balances the use of confident predictions with the exploration of less certain ones, preventing the model from becoming biased towards the nodes' most confident nodes. To mitigate, the noise of pseudo-labeling process, we also add pseudo-labels to minority classes and avoid the pseudo-labels for majority classes during training.

**Uncertainty-Aware:** Our proposed Selective Edge Removal (SER) method addresses the challenge of poor calibration in GNN pseudo-labeling by introducing a topology-aware uncertainty estimation approach. The method works by strategically removing edges based on node degrees, where highly-connected nodes have a higher probability of edge removal, then making multiple inferences on these perturbed graph versions. By measuring the entropy of predictions across different perturbed versions and computing the variance of these entropy values, we can effectively estimate prediction uncertainty for each unlabeled node. Nodes showing

consistent predictions (low entropy variance) across perturbations are considered more reliable and selected for pseudo-labeling, specifically retaining the lowest 90% based on uncertainty values. Rather than strictly attempting to separate aleatoric and epistemic uncertainty, this approach measures predictive stability under minor topological perturbations. To prevent severe topological shifts, we remove no more than 100 edges out of the thousands available in the graph. This micro-perturbation induces tiny variations in predictions; by measuring these variations and applying our thresholding mechanism, we can effectively filter out the most unstable nodes while preserving the graph's core structure. This provides a computationally efficient way to refine pseudo-label selection without strictly relying on standard entropy.

Finally, combining the thresholds for the pseudo-labeling process and the uncertainty estimation, we can select the pseudo-label for an unlabeled node, i.e.,

$$\tilde{Y}_i = \mathbb{1}(P(\hat{Y}_i|X_i) \geq \eta_l)\mathbb{1}(P(\hat{Y}_i|X_i) \leq \eta_u)\mathbb{1}\big(\mathrm{U}(P(\hat{Y}_i|X_i)) \leq Q_\alpha\big), \tag{3}$$

where $P(\hat{Y}_i|X_i)$ is the prediction of our model for node sample $X_i$ over classes and $\mathrm{U}(\cdot)$ is variance of entropy of each nodes predictions among different perturbations, $\mathrm{U}(P(\hat{Y}_i|X_i)) = \mathbb{VAR}_k(H_k(i))$ where $H_k(i)$ is the entropy of predictions for $i$-th node in the $k$-th perturbed graph. Also $Q_\alpha$ represent the quantile function for some $\alpha \in (0,1)$ as follows:

$$Q_\alpha = \mathrm{quantile}\big(\mathrm{U}(P(\hat{Y}_j|X_j)) : j = 1, \ldots, n, \alpha\big). \tag{4}$$

## 5.2 Class-Weighted Cross-Entropy loss function

To mitigate class imbalance during training, we use a class-weighted cross-entropy loss. The primary idea is to introduce a class-specific weighting factor, $\beta_c$, which adjusts the contribution of each class to the overall loss. This weighting factor is inversely proportional to the class frequency, assigning higher weights to underrepresented classes and lower weights to overrepresented classes. The formulation of the balanced softmax loss is given by

$$-\frac{1}{N}\sum_{i=1}^{N}\sum_{c=1}^{k}\beta_c y_{ic} \log(p_{ic}), \tag{5}$$

where $N$ is the number of training samples, $k$ is the number of classes, $y_{ic}$ is the ground truth label (0 or 1) for sample $i$ and class $c$, $p_{ic}$ is the predicted probability for sample $i$ and $c$-th class , and $\beta_c$ is the weighting factor calculated as $\beta_c = \frac{(1-\alpha)}{(1-\alpha^{m_c})}\alpha^{(1-m_c)}$. Here, $\alpha$ is a hyperparameter controlling the degree of balancing, and $m_c$ is the number of training samples belonging to $c$-th class. This adjustment ensures that the model pays more attention to the minority classes during training, improving performance on imbalanced datasets. Finally, the details and operational steps of the UPL algorithm are outlined in Appendix D.

Guided by our theoretical analysis, UPL introduces three novel mechanisms to specifically address the limitations of existing imbalanced learning methods:

1. **Topology-Aware Uncertainty (SER):** To ensure reliable pseudo-labels, we introduce Selective Edge Removal (SER). Rather than relying on simple softmax confidence, SER estimates uncertainty by observing prediction variance across topologically perturbed versions of the graph, capturing local structural stability.

2. **Band-Pass Pseudo-Label Filtering:** We employ a dual-threshold strategy (upper $\eta_u$ and lower $\eta_l$) that acts as a *band-pass* filter. This selects informative, mid-confidence nodes rather than just the *easiest* high-confidence samples, preventing the reinforcement of existing model biases.

3. **Minority-Class Augmentation:** Consistent with our risk bound findings, UPL exclusively targets unlabeled nodes predicted to belong to minority classes for pseudo-labeling. This directly mitigates the representation gap that dominates the generalization error.

Furthermore, our approach eliminates the need for computationally expensive clustering or node-reordering steps used in prior works like UNREAL, resulting in significant efficiency gains.

# 6 Experiment

In this section, we provide the results of experiments for our UPL algorithm. All experiments setup details are provided in Appendix E and G. All experiments are conducted on GCN models. More GNN architectures, e.g., GAT, GraphSage and GIN are studied in Appendix H.

**Baselines:** In our experiments, we compare our proposed method against a blend of both foundational and state-of-the-art (SOTA) methods. In particular, we include foundational baselines: Vanilla, Re-weight (Japkowicz & Stephen, 2002), DR-GCN (Shi et al., 2020) and BalancedSoftmax (Ren et al., 2020). Furthermore, we also consider recent SOTA methods, including, PC Softmax (Hong et al., 2021), GraphSMOTE (Zhao et al., 2021b), GraphENS (Park et al., 2022), TAM (Song et al., 2022), BAT1 (Liu et al., 2023), Graphsha (Li et al., 2023a), and Unreal (Yan et al., 2023).

**Datasets:** In our experiments, we utilize citation network datasets—CiteSeer, Cora, and Pubmed (Sen et al., 2008)—which we categorize as homophilic graphs, where nodes represent documents and edges represent citation links. For heterophilic graphs, we employ Chameleon and Squirrel (Rozemberczki et al., 2021), where nodes represent Wikipedia pages and edges denote links between them, as well as Wisconsin, a graph composed of web pages crawled from the Internet by Craven et al. (2000). Furthermore, we use the Amazon Computers and Amazon Photos datasets Shchur et al. (2019) in Appendix H.

**Imbalance Learning:** In a classification scenario with training dataset $z = \{z_1, \ldots, z_n\}$ where $z_i = (x_i, y_i)$, the imbalance occurs when the distribution $\mathbf{P}_k$ over $k$ classes within the dataset has larger probability mass points over some classes. The imbalance severity is often quantified using an imbalance ratio $\rho$, which measures the ratio between the number of samples in the most populated class (majority class) and the least populated class (minority class) and can be defined as follows,

$$\rho = \frac{\max_{j \in [k]} |m_j|}{\min_{j \in [k]} |m_j|}. \tag{6}$$

More information regarding Imbalance ratio definition and minority class discrimination can be found in Appendix G.3.

**Hyper-parameter Sensitivity**: We conducted a sensitivity analysis on the two thresholds used in pseudo-labeling to evaluate the model's performance at various values of $\eta_l$ and $\eta_u$. The results of this parameter sweep are presented in Figure 2.

The chart on the left shows the F1-score as $\eta_l$ is varied within the range $(0.25, 1.0)$, while $\eta_u$ is set to $\min(\eta_l + 0.3, 1.0)$. Based on this lower-threshold sweep, we observe that the optimal value of $\eta_l$ for this case lies within the range $(0.25, 0.45)$. A similar trend is observed across other datasets.

The right plot in Figure 2 depicts the results of varying $\eta_u$ within the range $(0.3, 1.0)$, with $\eta_l$ fixed at 0.3. These results further support the importance of using an upper threshold. Specifically, setting $\eta_u$ within the range $(0.35, 0.45)$ improves the overall F1-score on both the validation and test sets.

To rigorously evaluate the robustness of our band-pass pseudo-label filtering and ensure the thresholds are not merely heuristic, we expanded this analysis to a comprehensive two-dimensional hyperparameter sweep jointly varying both the lower threshold ($\eta_l$) and the upper threshold ($\eta_u$). Figure 3 presents a heatmap of the resulting F1 scores. To isolate the impact of the upper threshold, the heatmap colors are normalized for each fixed lower threshold. On this chart, the values on the antidiagonal represent experiments where the upper threshold is set to 1.0, effectively disabling the band-pass filter and reverting to standard high-confidence pseudo-labeling.

Analyzing the heatmap reveals that for any given lower threshold, applying an upper threshold ($\eta_u < 1.0$) improves the output to some degree. This substantiates the efficacy of the dual-threshold strategy in preventing the reinforcement of existing model biases. Furthermore, to verify the quality of the selected nodes, we tracked the minority-class recall and pseudo-label precision during these sweeps. We observed that pseudo-label precision peaked at $60.35\% \pm 4.32$ and minority-class recall reached $59.06\% \pm 4.59$ for Cora within our experiments, demonstrating that the dual-threshold approach successfully captures relevant minority samples.

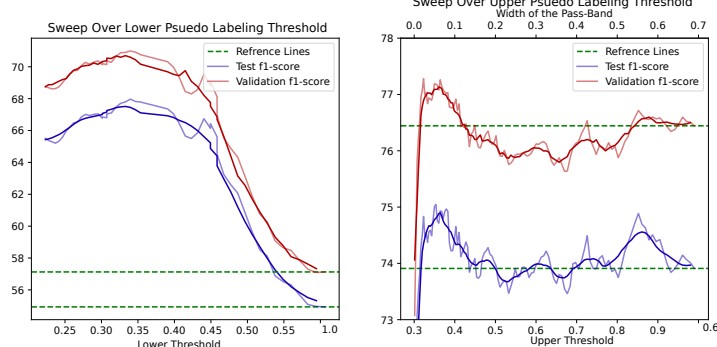

Figure 2: F1-score for different values of upper and lower thresholds for pseudo-labeling. The Figure on the left indicates the sweeping over $\eta_l$ while $\eta_u = 0.3 + \eta_l$ for the CiteSeer dataset. The Figure on the right represents sweeping over $\eta_u$ from 0.3 to 1.0 for Cora. In both Figures, the red line represents the validation results, and blue shows the result for the test set.

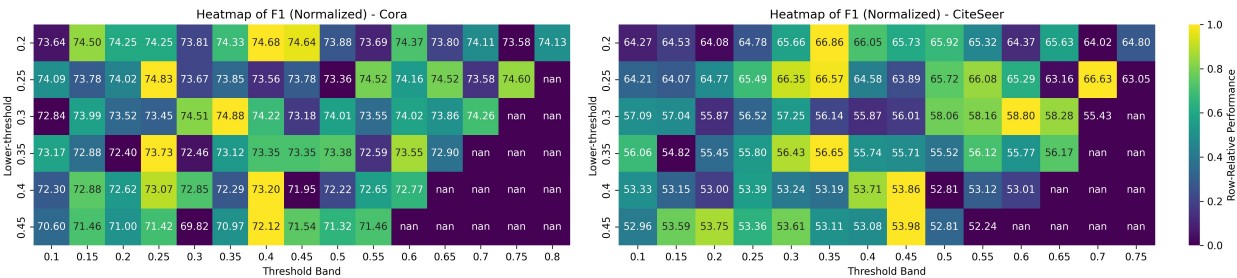

Figure 3: Heatmap displaying the F1 scores across different combinations of lower ($\eta_l$) and upper ($\eta_u$) thresholds. Colors are normalized per row to isolate the effect of $\eta_u$.

## 6.1 Experiment results

To further study the effect of UPL, we have conducted experiments on multiple datasets, a fraction of which has been shown in Table 1 and 2. We report averaged balanced accuracy (bAcc)[2] and F1-score with the standard errors for 10 repetitions on the GCN model. In our experiments, datasets are generated with an imbalance ratio of $\rho = 10$. We explore other imbalance ratios in Appendix H.

Table 1: Experimental results of our algorithm (UPL) compared to other baselines. All the experiments have been done using the GCN architecture. We report the averaged balanced accuracy, averaged F1-score and the standard error of each experiment for Cora, CiteSeer and PubMed datasets. All the results have been calculated over 10 repetitions and 10 pseudo-labeling iterations.

| Dataset | Cora | | CiteSeer | | PubMed | |
|---|---|---|---|---|---|---|
| **Imbalance Ratio** ($\rho = 10$) | bAcc. | F1 | bAcc. | F1 | bAcc. | F1 |
| Vanilla | $53.58\ _{\pm 0.70}$ | $48.52\ _{\pm 1.09}$ | $35.29\ _{\pm 0.34}$ | $22.62\ _{\pm 0.39}$ | $62.22\ _{\pm 0.35}$ | $48.99\ _{\pm 1.02}$ |
| Re-Weight(Japkowicz & Stephen, 2002) | $65.52\ _{\pm 0.84}$ | $65.54\ _{\pm 1.20}$ | $44.52\ _{\pm 1.22}$ | $38.85\ _{\pm 1.62}$ | $70.17\ _{\pm 1.25}$ | $66.37\ _{\pm 1.73}$ |
| PC Softmax Hong et al. (2021) | $67.79\ _{\pm 0.92}$ | $67.39\ _{\pm 1.08}$ | $49.81\ _{\pm 1.12}$ | $45.55\ _{\pm 1.26}$ | $70.20\ _{\pm 0.60}$ | $68.83\ _{\pm 0.73}$ |
| DR-GCN Shi et al. (2020) | $60.17\ _{\pm 0.83}$ | $59.31\ _{\pm 0.97}$ | $42.64\ _{\pm 0.75}$ | $38.22\ _{\pm 1.22}$ | $65.51\ _{\pm 0.81}$ | $64.95\ _{\pm 0.53}$ |
| GraphSMOTE Zhao et al. (2021b) | $62.66\ _{\pm 0.83}$ | $61.76\ _{\pm 0.96}$ | $34.26\ _{\pm 0.89}$ | $28.31\ _{\pm 1.48}$ | $68.94\ _{\pm 0.89}$ | $64.17\ _{\pm 1.43}$ |
| BalancedSoftmax Ren et al. (2020) | $67.75\ _{\pm 1.09}$ | $66.47\ _{\pm 1.27}$ | $52.09\ _{\pm 1.61}$ | $48.55\ _{\pm 2.01}$ | $71.78\ _{\pm 0.82}$ | $71.18\ _{\pm 0.94}$ |
| GraphENS Park et al. (2022)+ TAM Song et al. (2022) | $69.42\ _{\pm 0.90}$ | $68.70\ _{\pm 1.08}$ | $53.10\ _{\pm 1.82}$ | $50.67\ _{\pm 2.28}$ | $71.70\ _{\pm 0.95}$ | $69.01\ _{\pm 1.72}$ |
| GraphSHA Li et al. (2023a) | $39.82\ _{\pm 0.85}$ | $34.88\ _{\pm 0.92}$ | $33.20\ _{\pm 1.10}$ | $24.74\ _{\pm 1.25}$ | $66.55\ _{\pm 0.70}$ | $66.74\ _{\pm 0.68}$ |
| BAT1 Liu et al. (2023) | $72.32\ _{\pm 0.52}$ | $71.10\ _{\pm 0.55}$ | $59.23\ _{\pm 0.90}$ | $58.48\ _{\pm 0.87}$ | $73.37\ _{\pm 0.45}$ | $72.60\ _{\pm 0.48}$ |
| Unreal Yan et al. (2023) | $\mathbf{78.33}\ _{\pm 1.04}$ | $\mathbf{76.44}\ _{\pm 1.06}$ | $\underline{65.63}\ _{\pm 1.38}$ | $\underline{64.94}\ _{\pm 1.38}$ | $\underline{75.35}\ _{\pm 1.41}$ | $\underline{73.65}\ _{\pm 1.43}$ |
| UPL | $\underline{76.16}\ _{\pm 0.76}$ | $\underline{74.53}\ _{\pm 0.85}$ | $\mathbf{68.18}\ _{\pm 0.27}$ | $\mathbf{67.78}\ _{\pm 0.25}$ | $\mathbf{77.50}\ _{\pm 0.44}$ | $\mathbf{77.19}\ _{\pm 0.43}$ |

---

[2]Balanced accuracy is calculated as $\frac{1}{2}\left(\frac{\text{TP}}{\text{TP+FN}} + \frac{\text{TN}}{\text{TN+FP}}\right)$ where TP is true positive, FN is false negative and TN is true negative.

Table 2: Experimental results of our algorithm (UPL) compared to other baselines. All the experiments have been done using the GCN architecture. We report the averaged balanced accuracy, averaged F1-score and the standard error of each experiment for Cora, CiteSeer and PubMed datasets. All the results have been calculated over 10 repetitions and 10 pseudo-labeling iterations.

| Dataset | Chameleon | | Squirrel | | Wisconsin | |
|---|---|---|---|---|---|---|
| Imbalance Ratio ($\rho = 10$) | bAcc. | F1 | bAcc. | F1 | bAcc. | F1 |
| Vanilla | 57.01 ±0.53 | 55.61 ±0.67 | 38.07 ±0.57 | 34.81 ±0.69 | 34.51 ±2.31 | 31.45 ±2.66 |
| Re-Weight Japkowicz & Stephen (2002) | 36.07 ±0.87 | 35.61 ±0.81 | 26.92 ±0.53 | 25.04 ±0.59 | 44.13 ±3.08 | 40.74 ±3.27 |
| PC Softmax Hong et al. (2021) | 36.86 ±1.04 | 36.24 ±1.01 | 26.49 ±0.59 | 25.73 ±0.49 | 30.90 ±3.10 | 28.15 ±2.16 |
| DR-GCN Shi et al. (2020) | 33.34 ±0.81 | 29.60 ±0.79 | 23.34 ±0.43 | 18.20 ±0.49 | 29.44 ±1.36 | 27.08 ±1.37 |
| GraphSMOTE Zhao et al. (2021b) | 52.38 ±0.77 | 50.03 ±0.76 | 41.30 ±0.47 | 39.07 ±0.60 | 45.36 ±4.21 | 40.91 ±4.39 |
| BalancedSoftmax Ren et al. (2020) | 58.60 ±1.06 | 58.20 ±1.00 | 41.81 ±0.64 | 41.27 ±0.71 | 44.44 ±3.46 | 41.29 ±2.85 |
| GraphENS Park et al. (2022)+ TAM Song et al. (2022) | 40.98 ±0.85 | 38.93 ±0.84 | 27.98 ±0.90 | 24.04 ±0.70 | 41.38 ±2.78 | 34.56 ±2.22 |
| Unreal | 58.18 ±0.94 | 56.98 ±1.03 | **45.65** ±1.08 | **44.78** ±1.07 | 41.26 ±5.19 | 32.14 ±2.15 |
| **UPL** | **61.19** ±0.66 | **60.71** ±0.71 | 43.34 ±0.25 | 42.82 ±0.07 | **48.70** ±2.88 | **42.72** ±2.38 |

As shown in Tables 1 and 2, our UPL algorithm outperforms other baselines across both homophilic and heterophilic datasets in most cases. Notably, UPL demonstrates significantly lower performance variance than competing methods, indicating better reliability in its predictions. Table 2 also highlights a common challenge which is a marked decline in performance on heterophilic datasets. In contrast, our method delivers mostly better results in most cases, even in these challenging scenarios. Moreover, based on theoretical results and the fact that the maximum degree of nodes per class in heterophilic datasets is large, then we expect lower accuracy over these datasets. A detailed comparison with Unreal is provided in App. I.

Table 3: Experimental results of Ablation study of our proposed algorithm. All the experiments have been done using the GCN architecture. We report the averaged balanced accuracy, averaged F1-score and the standard error of each experiment for all datasets.

| Dataset | Cora | | CiteSeer | | PubMed | | Chameleon | | Squirrel | | Wisconsin | |
|---|---|---|---|---|---|---|---|---|---|---|---|---|
| Imbalance Ratio ($\rho = 10$) | bAcc. | F1 | bAcc. | F1 | bAcc. | F1 | bAcc. | F1 | bAcc. | F1 | bAcc. | F1 |
| Vanilla | 53.28 ±0.65 | 48.12 ±1.03 | 35.53 ±0.33 | 22.67 ±0.41 | 61.96 ±0.33 | 48.97 ±0.89 | 57.24 ±0.63 | 55.82 ±0.75 | 37.70 ±0.58 | 34.45 ±0.71 | 34.51 ±2.31 | 31.45 ±2.66 |
| + BS | 67.51 ±1.27 | 66.29 ±1.32 | 51.83 ±1.58 | 48.56 ±1.87 | 72.22 ±0.83 | 71.31 ±0.99 | 58.98 ±0.86 | 58.44 ±0.87 | 41.76 ±0.49 | 40.91 ±0.50 | 44.44 ±3.46 | 41.29 ±2.85 |
| + PL | 68.86 ±1.51 | 66.45 ±1.86 | 48.81 ±2.62 | 41.75 ±3.74 | 73.19 ±1.42 | 71.98 ±2.08 | 60.24 ±0.90 | 59.63 ±0.95 | 39.34 ±0.56 | 37.15 ±0.58 | 38.62 ±2.28 | 36.51 ±2.54 |
| + Uncertainty | 53.28 ±0.65 | 48.12 ±1.03 | 35.53 ±0.33 | 22.67 ±0.41 | 61.96 ±0.33 | 48.97 ±0.89 | 57.02 ±0.76 | 55.78 ±0.85 | 37.78 ±0.57 | 34.68 ±0.75 | 34.51 ±2.31 | 31.45 ±2.66 |
| + BS + PL | 75.27 ±0.92 | 73.59 ±1.02 | 64.70 ±1.17 | 64.34 ±1.26 | 76.08 ±0.28 | 76.08 ±0.29 | **61.26** ±0.73 | **60.73** ±0.74 | 41.83 ±0.42 | 41.04 ±0.48 | **47.72** ±4.08 | **44.03** ±3.44 |
| + Uncertainty + BS | 67.51 ±1.27 | 66.29 ±1.32 | 51.83 ±1.58 | 48.56 ±1.87 | 72.22 ±0.83 | 71.31 ±0.99 | 59.04 ±0.85 | 58.44 ±0.88 | 41.70 ±0.48 | 40.88 ±0.54 | 44.44 ±3.46 | 41.29 ±2.85 |
| + Uncertainty + PL | 67.51 ±2.16 | 65.28 ±2.59 | 40.01 ±2.01 | 28.25 ±2.31 | 74.85 ±0.75 | 73.80 ±0.89 | 59.21 ±0.74 | 58.14 ±0.77 | 38.39 ±0.67 | 35.86 ±0.87 | 36.92 ±2.62 | 34.74 ±2.76 |
| UPL | **76.16** ±0.76 | **74.53** ±0.85 | **68.18** ±0.27 | **67.78** ±0.25 | **77.50** ±0.44 | **77.19** ±0.43 | 61.19 ±0.66 | 60.71 ±0.71 | **42.34** ±0.67 | **41.76** ±0.68 | **48.70** ±2.88 | 42.72 ±2.38 |

## 6.2 Ablation study

We evaluate key components' individual and combined effects on our UPL algorithm to discover their contributions in transductive node classification tasks under imbalance scenario, for both homophilic and heterophilic datasets, in Table 3, respectively. For this purpose, we use Class-weighted CE (**CWCE**), which adjusts softmax to favor minority classes; Uncertainty-aware Pseudo-labeling (**UPL**), which uses generated pseudo-labeled samples to generate more training data; aiming to align predicted class distributions with actual distributions. In Tables 3, The Vanilla model serves as our baseline which is log-loss function. We further study other uncertainty method in App. H.

The results demonstrate the effectiveness of various strategies for addressing class imbalance in node classification tasks. Across most datasets, the UPL method excels compared to previous methods, showcasing the robustness of integrating approaches. Notably, combinations like CWCE + UPL frequently lead to significant performance improvements, highlighting the synergy between direct imbalance mitigation and the strategic use of pseudo-labeled data.

To specifically isolate the contribution of the topology-aware uncertainty estimator (Selective Edge Removal) from the broader pseudo-labeling mechanism, we conducted an extended ablation analysis. In this experiment, we held the lower threshold ($\eta_l$) constant while varying the upper threshold ($\eta_u$). Table 4 presents the performance differences across these threshold bands. The data confirms that applying the uncertainty yields improvements in most cases; across almost all threshold variations, the inclusion of uncertainty estimation enhances the F1 score compared to relying on baseline pseudo-labeling alone.

Table 4: Performance Differences by Threshold Band. This table demonstrates the effect of the uncertainty for varying $\eta_u$.

| $\eta_u$ | Cora | CiteSeer |
|------|------|----------|
| 0.3 | 0.48 | -0.76 |
| 0.4 | 1.00 | 0.02 |
| 0.5 | 0.97 | 1.46 |
| 0.6 | 2.38 | 2.22 |
| 0.7 | 1.63 | 2.72 |
| 0.8 | 1.37 | 0.28 |
| 0.9 | 1.31 | -0.32 |

## 7 Conclusion and Future Work

In this study, we make several key contributions to transductive node classification in imbalanced scenarios. First, we establish a theoretical upper bound on the population risk that depends on the maximum degree node within each class. Second, we develop an uncertainty-aware algorithm for identifying pseudo-labels for minority class nodes, introducing a novel selective edge removal method for uncertainty estimation. Our approach demonstrates top-tier performance compared to baseline methods across both homophilic and heterophilic datasets.

While our theoretical analysis currently focuses on $\gamma$-margin loss functions in binary classification, several promising directions emerge for future work. From a theoretical perspective, we aim to extend our results to inductive node classification settings where unlabeled nodes are not observed during training. Additionally, we plan to investigate theoretical bounds for heterophilic graphs to strengthen our framework's theoretical foundation.

As future work, we envision integrating our approach with H2GCN approach (Zhu et al., 2020) to enhance performance on heterophilic datasets. We also plan to incorporate advanced pseudo-labeling techniques such as Meta-Pseudo Label (Pham et al., 2021) or Flexmatch (Zhang et al., 2021a) to improve label quality. Finally, we intend to explore subgraph-based selective edge removal approaches for more sophisticated uncertainty estimation.

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

## A  Notations

Notations in this paper are summarized in Table 5.

Table 5: Summary of notations in the paper

| Notation | Definition | Notation | Definition |
|---|---|---|---|
| $\mathcal{A}$ | Adjacency matrices space | $X_i$ | $i$-th node feature |
| $\deg_i^{\max}$ | Maximum node degree i-th class | $\deg_i^{\min}$ | Minimum node degree i-th class |
| $Y_i$ | $i$-th node's label | $Z_i$ | $i$-th node sample pair $(X_i, Y_i)$ |
| $\mathbf{Z}_m$ | The set of data training samples | $\mathbf{Z}_u$ | The unlabeled samples |
| $P(.)$ | Prediction of model | $m_i$ | Number of nodes in $i$-th class |
| $k$ | Number of classes | $P_j$ | $\frac{|m_j|}{\sum_{i=1}^{k}|m_i|}$ |
| $d_{\min}$ | Minimum node degree of among all classes | $G_f(\mathbf{A})$ | Graph filter with input matrix $\mathbf{A}$ |
| $l(\hat{y}, y)$ | Loss function | $h_\theta$ | Parameterized model |
| $B_f$ | $l_2$-ball of radius | $\theta_i$ | Parameter matrix of i-th layer |
| $\varphi$ | Activation function | $\mathfrak{R}_{m+u}(\mathcal{H})$ | Empirical Rademacher complexity |
| $R(Z_u, h_\theta)$ | True risk over unlabeled dataset | $R(Z_m, h_\theta)$ | Empirical risk over labeled dataset |
| $U_F(i)$ | Maximum of Frobenius norm i-th layer parameters | $U(.)$ | Uncertainty Function |
| $\eta_l$ | Lower thresholds of Pseudo-labeling process | $\eta_u$ | Upper thresholds of Pseudo-labeling process |
| $KL(P\|Q)$ | $\int_{\mathbb{Z}} \log(\mathrm{d}P/\mathrm{d}Q)\mathrm{d}P$ | $H(P, Q)$ | $\int_{\mathbb{Z}} \log(\mathrm{d}P)\mathrm{d}Q$ |
| $B_f$ | Bound on node features | $\rho$ | Imbalance Ratio |
| $\|\mathbf{Y}\|_F$ | $\sqrt{\sum_{j=1}^{k}\sum_{i=1}^{q}\mathbf{Y}^2[j,i]}$ | $\|\mathbf{Y}\|_\infty$ | $\max_{1\le j\le k}\sum_{i=1}^{q}|\mathbf{Y}[j,i]|$ |
| $\|\mathbf{Y}\|_2$ | $\left(\sum_{j=1}^{q}|q|x_j^2\right)^{1/2}$ | $Q_\alpha$ | Quantile of order $\alpha$ |
| $t$ | Number of itterations | $S_k$ | Number of edges removed in each itteration |

## B  Other Related Works

**Pseudo-labeling in semi-supervised learning:** Pseudo-labeling, introduced in (Lee et al., 2013), involves training a model using labeled data and assigning pseudo-labels to the unlabeled data based on the model's predictions. These pseudo-labels are then used to construct another model, which is trained in a supervised manner using both labeled and pseudo-labeled data. Under different scenarios, it is shown that the pseudo-labeling is effective, (Arazo et al., 2020) and (Rizve et al., 2021). (Kou et al., 2023) shows that semi-supervised learning with pseudo-labeling can achieve near-zero test loss under some conditions. The study by Pham et al. (2021) introduced meta pseudo-labeling. This method enhanced pseudo-labels' accuracy by incorporating feedback from the student model. (Rizve et al., 2021) proposed confidential-based pseudo-label generation for training a network with unlabeled data. (Arazo et al., 2020) suggests soft-labeling with the MixUp method to reduce over-fitting to model predictions and confirmation bias. In contrast, our work is focused on transductive node classification, and we employed the Pseudo-labels to mitigate the imbalance effect.

**Imbalance Classification:** Effective classification in the presence of imbalanced data is an important research field in supervised and semi-supervised learning, given that a significant class imbalance is commonly found in numerous real-world contexts (Herland et al., 2018; Rao et al., 2006; Cieslak et al., 2006) such as fraud detection in finance applications (Wei et al., 2013). This imbalance often presents challenges, as many classifiers tend to prefer the majority class, sometimes to the extent of completely overlooking the minority class. Traditional Class Imbalance Learning methods typically fall into two broad categories: data-level methods and algorithm-level methods. Data-level techniques aim to rebalance the training data itself to mitigate class imbalance. This includes strategies like over-sampling the minority class (Chawla et al., 2002; He et al., 2008; Mullick et al., 2019), under-sampling the majority class (Laurikkala, 2001; Mani & Zhang, 2003), and hybrid methods that combine both over-sampling and under-sampling (Ramentol et al., 2012; Sáez et al., 2015). On the other hand, algorithm-level methods focus on modifying the learning algorithms to better handle class imbalance. These methods encompass cost-sensitive learning (Tang et al., 2008; Zhou & Liu, 2005), ensemble learning (Seiffert et al., 2009; Guo & Viktor, 2004; Chawla et al., 2003), and engineering of specialized loss functions (Cao et al., 2019; Dong et al., 2018; Cui et al., 2019).

# C   Proof and details of Section 4

**Lemma C.1** (Bound on infinite norm of the symmetric normalized graph filter). *Consider a graph sample $G$ with adjacency matrix $A$. For the symmetric normalized graph filter, we have $\left\|\tilde{D}^{-1/2}(A+I)\tilde{D}^{-1/2}\right\|_\infty[k] \leq \sqrt{\frac{\deg_k^{\max}+1}{\deg^{\min}+1}}$ where $\deg_k^{\max}$ is the maximum degree of graph $G$ with adjacency matrix $A$ among the nodes in $k$-th class.*

*Proof.* Recall that $\tilde{A} = A + I$. We have,

$$
\begin{aligned}
\left\|\tilde{L}\right\|_\infty[k] &= \left\|\tilde{D}^{-1/2}\tilde{A}\tilde{D}^{-1/2}\right\|_\infty[k] \\
&= \max_{i\in[N_k]} \sum_{j=1}^N \frac{\tilde{A}_{ij}}{\sqrt{\deg_i+1}\sqrt{\deg_j+1}} \\
&\leq \frac{1}{\sqrt{\deg^{\min}+1}} \max_{i\in[N_k]} \sum_{j=1}^N \frac{\tilde{A}_{ij}}{\sqrt{\deg_i+1}} \\
&\leq \sqrt{\frac{\deg_k^{\max}+1}{\deg^{\min}+1}}.
\end{aligned}
\tag{7}
$$

where $N_k$ is the number nodes in $k$-th class. $\qquad\square$

**Proposition 4.4** (**Restated**). *Let $Q_i = (\frac{1}{u_i} + \frac{1}{m_i})$, $S_i = \frac{m_i+u_i}{(m_i+u_i-1/2)(1-1/(2\max(m_i,u_i)))}$ for $i = 1, 2$. Then, with probability at least $(1-\delta/2)$ over the choice of the training set from nodes of graphs, for all $h_\theta \in \mathcal{H}$,*

$$
R(\mathbf{Z}_u, h_\theta) \leq \sum_{i=1}^2 R_\gamma(\mathbf{Z}_{m_i}, h_\theta) + \frac{u_i}{u}\left(\frac{1}{\gamma}\mathfrak{R}_{m_i+u_i}(\mathcal{H}) + c_0 Q_i \sqrt{\min(m_i, u_i)} + \sqrt{\frac{S_i Q_i}{2}\log(1/\delta)}\right),
$$

*where $\mathfrak{R}_{m_i+u_i}(\mathcal{H})$ is the transductive Rademacher complexity for $i$-th class, $R_\gamma(\mathbf{Z}_{m_i}, h_\theta)$ is the empirical risk based on $\gamma$-margin loss for $i$-th class and $c_0 = \sqrt{\frac{32\ln(4e)}{3}}$.*

*Proof.* Let's decompose the true risk as follows,

$$
R(\mathbf{Z}_u, h_\theta) = \sum_{i=1}^2 \frac{u_i}{u} R(\mathbf{Z}_{u_i}, h_\theta),
\tag{8}
$$

where $R(\mathbf{Z}_{u_i}, h_\theta)$ is true risk of $i$-th class. Inspired by (El-Yaniv & Pechyony, 2009, Corollary 1), we have,

$$
R(\mathbf{Z}_{u_i}, h_\theta) \leq R_\gamma(\mathbf{Z}_{m_i}, h_\theta) + \frac{1}{\gamma}\mathfrak{R}_{m_i+u_i}(\mathcal{H}) + c_0 Q_i \sqrt{\min(m_i, u_i)} + \sqrt{\frac{S_i Q_i}{2}\log(1/\delta)}.
\tag{9}
$$

Combining equation 9 with equation 8, we have,

$$
\begin{aligned}
R(\mathbf{Z}_u, h_\theta) &= \sum_{i=1}^2 \frac{u_i}{u} R(\mathbf{Z}_{u_i}, h_\theta) \\
&\leq \sum_{i=1}^2 \frac{u_i}{u}\left(R_\gamma(\mathbf{Z}_{m_i}, h_\theta) + \frac{1}{\gamma}\mathfrak{R}_{m_i+u_i}(\mathcal{H}) + c_0 Q_i \sqrt{\min(m_i, u_i)} + \sqrt{\frac{S_i Q_i}{2}\log(1/\delta)}\right).
\end{aligned}
\tag{10}
$$

It completes the proof by considering $\frac{u_i}{u} \leq 1$ and also using union bound. $\qquad\square$

**Proposition 4.5** (**Restated**). *Given Assumptions 4.1 , 4.2 and 4.3, then the following upper bound holds on the transductive Rademacher complexity of node classification,*

$$
\mathfrak{R}_{m_k+u_k}(\mathcal{H}) \leq B_f(\sqrt{2\log(2)d}+1)\Pi_{j=1}^d U_F(i)\|\mathrm{G}_f(\mathbf{A})[i]\|_\infty^{2(d-1)}\sqrt{\frac{1}{m_k} + \frac{1}{u_k}}
$$

*where $\|\mathrm{G}_f(\mathbf{A})[k]\|_\infty$ is the infinite norm of graph filter among $k$-th class.*

*Proof.* Fix $\lambda > 0$, to be chosen later. Let us denote the output of $q$-th layer of GCN as,

$$N_{\theta_q}(\mathbf{X}_{u_k+m_k}) = \sigma(\theta_{q-1}\mathrm{G}_f(\mathbf{A})[i,:](\cdots\sigma(\theta_1\mathrm{G}_f(\mathbf{A})[i,:]\mathbf{X}_{u_k+m_k}))).$$

For the last layer, we do not consider aggregation. The Rademacher complexity can be upper bounded as

$$\left(\frac{m_k u_k}{m_k + u_k}\right)\mathfrak{R}_{m_k+u_k}(\mathcal{H}) = \mathbb{E}_{\boldsymbol{\epsilon}}\sup_{N_{\theta_{d-1}},\theta_d}\sum_{i=1}^{m_k+u_k}\epsilon_i\theta_d\sigma(\theta_{d-1}\mathrm{G}_f(\mathbf{A})[i,:]N_{\theta_{d-1}}(\mathbf{X}_{u_k+m_k}))$$

$$\leq \frac{1}{\lambda}\log\mathbb{E}_{\boldsymbol{\epsilon}}\sup\exp\left(\lambda\sum_{i=1}^{m_k+u_k}\epsilon_i\theta_d\sigma(\theta_{d-1}\mathrm{G}_f(\mathbf{A})[i,:]N_{\theta_{d-1}}(\mathbf{X}_{u_k+m_k}))\right)$$

$$\leq \frac{1}{\lambda}\log\mathbb{E}_{\boldsymbol{\epsilon}}\sup\exp\left(U_F(d)\cdot\left\|\lambda\sum_{i=1}^{m_k+u_k}\epsilon_i\sigma(\theta_{d-1}\mathrm{G}_f(\mathbf{A})[i,:]N_{\theta_{d-1}}(\mathbf{X}_{u_k+m_k}))\right\|\right).$$

We write this last expression as

$$\frac{1}{\lambda}\log\mathbb{E}_{\boldsymbol{\epsilon}}\sup_{f,\|\theta_{d-1}\|_F\leq U_F(d-1)}\exp\left(U_F(d)\cdot\lambda\left\|\sum_{i=1}^{m_k+u_k}\epsilon_i\sigma(\theta_{d-1}\mathrm{G}_f(\mathbf{A})[i,:]N_{\theta_{d-1}}(\mathbf{X}_{u_k+m_k}))\right\|\right)$$

$$\leq \frac{1}{\lambda}\log\left(2\cdot\mathbb{E}_{\boldsymbol{\epsilon}}\sup_f\exp\left(U_F(d)\cdot U_F(d-1)\cdot\|\mathrm{G}_f(\mathbf{A})\|_\infty^2\lambda\left\|\sum_{i=1}^{m_k+u_k}\epsilon_iN_{\theta_{d-1}}(\mathbf{X}_{u_k+m_k})\right\|\right)\right).$$

Repeating the process, we arrive at

$$\left(\frac{m_k u_k}{m_k + u_k}\right)\hat{\mathcal{R}}_{m_i+u_i}(\mathcal{H}) \leq \frac{1}{\lambda}\log\left(2^d\cdot\mathbb{E}_{\boldsymbol{\epsilon}}\exp\left(M\lambda\left\|\sum_{i=1}^{m_k+u_k}\epsilon_i\mathbf{X}_i\right\|\right)\right). \tag{11}$$

where $V = \|\mathrm{G}_f(\mathbf{A})\|_\infty^{2(d-1)}\prod_{j=1}^d U_F(j)$. Define a random variable

$$Z = V\cdot\left\|\sum_{i=1}^m\epsilon_i\mathbf{X}_i\right\|,$$

(random as a function of the random variables $\epsilon_1,\ldots,\epsilon_m$). Then

$$\frac{1}{\lambda}\log\left\{2^d\cdot\mathbb{E}\exp\lambda Z\right\} = \frac{d\log(2)}{\lambda} + \frac{1}{\lambda}\log\left\{\mathbb{E}\exp\lambda(Z-\mathbb{E}Z)\right\} + \mathbb{E}Z. \tag{12}$$

By Jensen's inequality, $\mathbb{E}[Z]$ can be upper bounded by using the Rademacher complexity random variables

$$V\sqrt{\mathbb{E}_{\boldsymbol{\epsilon}}\left[\left\|\sum_{i=1}^m\epsilon_i\mathbf{X}_i\right\|^2\right]} = V\sqrt{\mathbb{E}_{\boldsymbol{\epsilon}}\left[\sum_{i,i'=1}^m\epsilon_i\epsilon_{i'}\mathbf{X}_i^\top\mathbf{x}_{i'}\right]} = V\sqrt{2p\sum_{i=1}^m\|\mathbf{X}_i\|^2},$$

where $p = \frac{m_k u_k}{(m_k + u_k)^2}$. To handle the $\log\left\{\mathbb{E}\exp\lambda(Z-\mathbb{E}Z)\right\}$ term in equation 12, note that $Z$ is a deterministic function of the i.i.d. random variables $\epsilon_1,\ldots,\epsilon_m$, and satisfies

$$Z(\epsilon_1,\ldots,\epsilon_i,\ldots,\epsilon_m) - Z(\epsilon_1,\ldots,-\epsilon_i,\ldots,\epsilon_m) \leq 2V\|\mathbf{X}_i\|.$$

This means that $Z$ satisfies a bounded-difference condition, which by the proof of Theorem 6.2 in (Boucheron et al., 2003), implies that $Z$ is sub-Gaussian, with variance factor

$$v = \frac{1}{4}\sum_{i=1}^{m_k+u_k}(2V\|\mathbf{X}_i\|)^2 = 2pV^2\sum_{i=1}^{m_k+u_k}\|\mathbf{X}_i\|^2,$$

and satisfies

$$\frac{1}{\lambda} \log \left\{ \mathbb{E} \exp \lambda(Z - \mathbb{E}Z) \right\} \leq \frac{1}{\lambda} \frac{\lambda^2 2V^2 p \sum_{i=1}^{m_k+u_k} \|\mathbf{X}_i\|^2}{2} = \frac{\lambda 2V^2 p \sum_{i=1}^{m_k+u_k} \|\mathbf{X}_i\|^2}{2}.$$

Choosing $\lambda = \dfrac{\sqrt{2\log(2)d}}{2pV\sqrt{\sum_{i=1}^{m_k+u_k} \|\mathbf{X}_i\|^2}}$ and using the above, we get that equation 11 can be upper bounded as follows:

$$\frac{1}{\lambda} \log \left\{ 2^d \cdot \mathbb{E} \exp \lambda Z \right\} \leq \mathbb{E}Z + \sqrt{2\log(2)d} \cdot V \sqrt{2p \sum_{i=1}^{m_k+u_k} \|\mathbf{X}_i\|^2}$$

$$\leq V\left(\sqrt{2\log(2)d}+1\right) \sqrt{2p \sum_{i=1}^{m_k+u_k} \|\mathbf{X}_i\|^2}$$

$$\leq \sqrt{2p}V\left(\sqrt{2\log(2)d}+1\right) B_f \sqrt{m_k + u_k}$$

$$= \sqrt{2p}V\left(\sqrt{2\log(2)d}+1\right) B_f \sqrt{m_k + u_k},$$

from which the result follows. □

**Theorem 4.6 (Restated).** *Under the same Assumptions in Proposition 4.4 and Proposition 4.5, the following upper bound holds on population risk for imbalanced transductive node classification under a GCN model,*

$$R(\mathbf{Z}_u, h_\theta) \leq \sum_{i=1}^{2} R_\gamma(\mathbf{Z}_{m_i}, h_\theta) + \frac{u_i}{u}\Bigg(\frac{B_f(\sqrt{2\log(2)d}+1)\Pi_{j=1}^d U_F(j)}{\gamma} \sqrt{\frac{1}{m_i} + \frac{1}{u_i}} \times \left(\frac{\deg_i^{\max}+1}{\deg^{\min}+1}\right)^{d-1}$$

$$+ c_0 Q_i \sqrt{\min(m_i, u_i)} + \sqrt{\frac{S_i Q_i}{2}\log(1/\delta)}\Bigg),$$

*where $U_F(i)$ is the maximum of Frobenius norm of $i$-th layer parameter.*

*Proof.* The proof follows directly from combining Proposition 4.4 with Proposition 4.5 and substituting $\|\mathbf{G}_f(\mathbf{A})\|_\infty$ for the GCN model using Lemma C.1. □

# D   UPL Algorithm Details

---
**Algorithm 1** Uncertainty-aware Pseudo-labeling (UPL) Algorithm

---
1: **Input:** Graph $G(X, V, E, Y)$, set of known train labels $\mathcal{Y}_t$, Number of classes $C$, set of edges $\mathcal{E}$, Number of Iterations $k$
2: **Output:** Model parameters
3: Initialize empty pseudo labels set $\mathcal{Y}_p$
4: **for** 1 to $k$ **do**
5:  Initialize new model
6:  Train *model* on $\mathcal{Y}_t \cup \mathcal{Y}_p$ utilizing CWCE
7:  Initialize results set $R$ with known labels
8:  **for** $i \leftarrow 1$ to $t$ **do**
9:   $G_{\text{Aug}} \leftarrow$ Selective Edge Removal on $G$ for $S_k$ times
10:   $R_i \leftarrow \{\text{Inference Model on } G_{\text{Aug}}\}$          ▷ Add output to $R$
11:  **end for**
12:  $P \leftarrow \{\text{Inference Model on } G\}$            ▷ class-wise probabilities
13:  Initialize the set $\mathcal{Y}^{\text{new}}$ for new pseudo labels
14:  **for** each $v$ in $G(V)$ **do**
15:   **if** $v \notin \mathcal{Y}_t$ **then**
16:    **if** $\eta_l < \max(P_k^v) < \eta_u$ **then**
17:     **if** $U(v) < Q_\alpha$ **then**
18:      Add $v$ and $\arg\max(P_k^v)$ to $\mathcal{Y}^{\text{new}}$ as node and pseudo label
19:     **end if**
20:    **end if**
21:   **end if**
22:  **end for**
23:  $\mathcal{Y}^p \leftarrow \mathcal{Y}^{\text{new}}$
24: **end for**
25: **return** *best_model* parameters

---

# E   Experiment Details

**Datasets:** We summarized the dataset statistics in Table 6. Note that in the Amazon Computers and Amazon Photos datasets Shchur et al. (2019), nodes represent products and edges denote co-purchasing relationships between them. We have included the Flickr dataset Zeng et al. (2019) to demonstrate our model's performance among the large datasets.

Table 6: Dataset statistics.

| Dataset | Nodes | Edges | Classes | Features |
|---|---|---|---|---|
| CiteSeer | 3,327 | 4,732 | 6 | 3,703 |
| Cora | 2,708 | 5,429 | 7 | 1,433 |
| Pubmed | 19,717 | 44,338 | 3 | 500 |
| Chameleon | 2,277 | 36,101 | 5 | 2,325 |
| Squirrel | 5,201 | 217,073 | 5 | 2,089 |
| Wisconsin | 251 | 515 | 5 | 1,703 |
| Computers | 13,381 | 245,778 | 10 | 767 |
| Photo | 7,650 | 119,081 | 8 | 745 |
| Flickr | 89,250 | 899,756 | 7 | 500 |

**Hyperparameters:** To attain the best results we have fine-tuned three hyperparameters for our model. These hyperparameters and their tune range along with hyperparameters from some of the previous have been displayed in Table 7.

Table 7: Tuned hyperparameters used in our work compared with those from previous methods

| Methods | Hyperparameters Range |
|---|---|
| GraphENS | $\lambda \in \beta(2,2)$ 
 $k \in \{1,5,10\}$ 
 $\tau \in \{1,2\}$ 
 warm up $\in \{1,5\}$ |
| TAM | $\alpha \in \{0.25, 0.5, 1.5, 2.5\}$ 
 $\beta \in \{0.125, 0.25, 0.5\}$ 
 $\phi \in \{0.8, 1.2\}$ |
| Unreal | $K \in \{100, 300, 500, 700, 900\}$ 
 $T \in \{40, 50, 60, 70, 80, 100\}$ 
 $\alpha \in \{4, 6, 8, 10\}$ 
 $p \in \{0.5, 0.75, 0.98\}$ 
 DGIN threshold $\in \{0.25, 0.5, 0.75, 1.00\}$ |
| UPL | $\eta_l \in \{0.25 + 0.05n \mid n \leq 9, n \in \mathbb{N} \cup \{0\}\}$ 
 $\eta_u \in \{min(\eta_l + 0.25n, 1.00) \mid n \leq 3, n \in \mathbb{N}\}$ 
 $Q_\alpha \in \{Q_{0.8}, Q_{0.9}\}$ |

## F   Empirical Evaluation of Uncertainty and Pseudo-Labeling

To better illustrate the dynamics of the Selective Edge Removal (SER) component and to provide a more granular evaluation of our pseudo-labeling strategy, we tracked the pseudo-label precision and minority class recall across five distinct experimental runs on the Cora and CiteSeer datasets. The results are summarized in Table 8.

Table 8: Pseudo-label precision and minority class recall across five distinct experimental runs for the Cora and CiteSeer datasets. These results demonstrate the exploratory nature of the SER mechanism.

| Dataset | Metric (%) | Run 1 | Run 2 | Run 3 | Run 4 | Run 5 | Average |
|---|---|---|---|---|---|---|---|
| Cora | Pseudo-Label Precision | 88.24 | 73.81 | 35.42 | 55.17 | 51.52 | **60.83** |
| | Minority Class Recall | 42.19 | 67.19 | 67.19 | 60.94 | 57.81 | **59.06** |
| CiteSeer | Pseudo-Label Precision | 72.81 | 68.22 | 47.06 | 74.81 | 59.22 | **64.42** |
| | Minority Class Recall | 38.96 | 36.36 | 35.06 | 41.56 | 37.66 | **37.92** |

While maximizing pseudo-label accuracy is a common objective in self-training, we emphasize that the primary goal of our SER-based self-learning process is to balance exploitation with exploration. The reported minority class recall metrics highlight this design philosophy. By intentionally allowing nodes with lower-confidence predictions to enter the training loop, SER encourages the exploration of difficult, underrepresented classes, effectively preventing the model from collapsing into highly confident but biased representations of the majority class. Because SER intentionally injects this controlled noise to facilitate exploration, traditional uncertainty-based error-detection metrics (such as AUC or calibration curves) may interpret this intended exploratory behavior as a limitation, whereas it functions as a regularizer against the reinforcement of majority-class bias.

## G   Experiments Setup

UPL contains two tunable hyperparameters used for the pseudo labels threshold and three fixed ones which have approximated using a few experiments. Other extra hyperparameters have been adjusted according to (Song et al., 2022). We will discuss the setting for these parameters in the following sections.

## G.1 UPL

We employ two thresholds for pseudo-labeling. As mentioned in Table 7, $\eta_l$ and $\eta_u$ are chosen respectively from $\{0.25 + 0.05n \mid 0 \le n \le 8, n \in \mathbb{N}\}$ and $\eta_u \in \{min(\eta_l + 0.25n, 1) \mid n \le 3, n \in \mathbb{N}\}$. Both $S_k$ and $t$ are fixed to 100, and the quantile for uncertainty-based node selection is chosen from $\{Q_{0.8}, Q_{0.9}\}$.

## G.2 Class-weighted CE

Another part of UPL is Class-weighted CE (CWCE) presented in (Ren et al., 2020). We have used the same values found in (Song et al., 2022) to prevent increasing the number of hyperparameters in the UPL algorithm.

## G.3 Imbalance Ratio and Class Discrimination

In our experiments, the distinction between minority and majority classes is determined from the labeled training set. The **imbalance ratio** ($\rho$) is formally defined as the ratio of the size of the largest class to that of the smallest class. This is calculated as:

$$\rho = \frac{\max_{j \in [k]} |m_j|}{\min_{j \in [k]} |m_j|} \tag{13}$$

where $|m_j|$ is the number of training samples in class $j$.

Following this, we define a "minority class" as any class whose size is approximately $\frac{|m_{\max}|}{\rho}$, where $|m_{\max}|$ is the size of the largest class. In our UPL framework, we exclusively add pseudo-labels for nodes that the model predicts to belong to one of these identified minority classes.

We also note that for certain datasets like **Wisconsin**, which are naturally imbalanced and where training set size may vary significantly across different experimental runs, we take an additional step for robustness. To establish a stable imbalance ratio, we first calculate the average size of each class across all experimental runs and then apply the formula above to these averaged sizes. This ensures a consistent and representative definition of minority and majority classes throughout our experiments.

**Imbalance Ratio:** The distribution of training sets for each dataset can be seen in Table 9. All datasets follow a 10:1 imbalance ratio, except for Wisconsin, which has an imbalance ratio ($\rho$) of 11.63. By analyzing Table 9, it is evident from the results in Tables 11 and 12 that our model generally performs significantly better when the node with the maximum degree is present in one of the majority classes.

## G.4 Other Details

More minor details concerning the experiments have been explained in the following sections.

Table 9: Number of training nodes per class

| Dataset ($\rho = 10$) | $\mathbf{C_0}$ | $\mathbf{C_1}$ | $\mathbf{C_2}$ | $\mathbf{C_3}$ | $\mathbf{C_4}$ | $\mathbf{C_5}$ | $\mathbf{C_6}$ | $\mathbf{C_7}$ | $\mathbf{C_8}$ | $\mathbf{C_9}$ |
|---|---|---|---|---|---|---|---|---|---|---|
| Cora | 20 | 20 | 20 | 20 | 2 | 2 | 2 | - | - | - |
| CiteSeer | 20 | 20 | 20 | 2 | 2 | 2 | - | - | - | - |
| PubMed | 20 | 20 | 2 | - | - | - | - | - | - | - |
| Chameleon | 225 | 220 | 218 | 22 | 22 | - | - | - | - | - |
| Squirrel | 487 | 494 | 501 | 50 | 50 | - | - | - | - | - |
| Wisconsin ($\rho = 11.63$) | 4 | 38 | 50 | 5 | 5 | - | - | - | - | - |
| Computers | 20 | 20 | 20 | 20 | 20 | 2 | 2 | 2 | 2 | 2 |
| Photo | 20 | 20 | 20 | 20 | 2 | 2 | 2 | 2 | - | - |
| Flickr | 2628 | 4321 | 3164 | 2431 | 432 | 432 | 432 | - | - | - |

**Hardware and setup:** All of our Experiments were conducted on a single server containing an Nvidia RTX 3090 GPU, AMD Ryzen 5 3600x CPU @ 3.80GHz, and 48GB of RAM.

Table 10: Size of maximum degree node per class

| Dataset | $C_0$ | $C_1$ | $C_2$ | $C_3$ | $C_4$ | $C_5$ | $C_6$ |
|---|---|---|---|---|---|---|---|
| Cora | 36 | 78 | 168 | 74 | 40 | 23 | 31 |
| CiteSeer | 15 | 21 | 99 | 18 | 51 | 17 | - |
| PubMed | 80 | 171 | 154 | - | - | - | - |
| Chameleon | 207 | 269 | 322 | 531 | 732 | - | - |
| Squirrel | 1041 | 1054 | 1081 | 1060 | 1904 | - | - |
| Wisconsin | 5 | 122 | 10 | 13 | 11 | - | - |

**Training:** To choose the training mask, we use the training masks provided by Pytorch Geometric. We train each model for 1000 epochs with an early stopping patience of 100. For each dataset $\eta_l$ is chosen from $(0.25, 0.90)$ and the value of $\eta_u$ is chosen from $(\eta_u, 1.00)$. We also perform each experiment with 10 iterations of pseudo-labeling.

**Optimizer:** we use PyTorch Adaptive Moment Estimation (ADAM) as the optimizer. Also, $\ell_2$ regularization with weight decay of $5e^{-4}$ for the first layer and dropout in some layers are used to prevent over-fitting. We use a learning rate of 0.01.

# H   More Experiments

## H.1   Other Imbalance Ratios

In this section, we present the performance outcomes for four prominent GNN architectures: Graph Convolutional Network (GCN), Graph Attention Network (GAT), Graph SAGE (SAGE), and Graph Isomorphism Network (GIN). These models were evaluated across six diverse datasets with four different imbalance ratios to assess their effectiveness in various scenarios. The results for the imbalance ratio of 10 are detailed in Table 11 and Table 12.

Tables 13 and 14 present the performance evaluation of UPL with an imbalance ratio of 5, while Tables 15 and 16 show the corresponding results for a lower imbalance ratio of $\rho = 2$.

We have also assessed our performance under an imbalance ratio of 20. These results are presented in Tables 17 and 18 for the Homophilic and Heterophilic datasets, respectively.

Table 11: Experimental results of our regularization methods and other baselines on three class-imbalanced node classification benchmark datasets: Cora, CiteSeer and PubMed. We report averaged balanced accuracy (bAcc.) and F1-score with the standard errors for 10 repetitions on three representative GNN architectures, i.e., GCN, GAT, Sage and GIN.

| | Dataset | Cora | | CiteSeer | | PubMed | |
|---|---|---|---|---|---|---|---|
| | **Imbalance Ratio ($\rho = 10$)** | bAcc. | F1 | bAcc. | F1 | bAcc. | F1 |
| GCN | Vanilla | 53.58 ±0.70 | 48.52 ±1.09 | 35.29 ±0.34 | 22.62 ±0.39 | 62.22 ±0.35 | 48.99 ±1.02 |
| | Re-Weight | 65.52 ±0.84 | 65.54 ±1.20 | 44.52 ±1.22 | 38.85 ±1.62 | 70.17 ±1.25 | 66.37 ±1.73 |
| | PC Softmax | 67.79 ±0.92 | 67.39 ±1.08 | 49.81 ±1.12 | 45.55 ±1.26 | 70.20 ±0.60 | 68.83 ±0.73 |
| | DR-GCN | 60.17 ±0.83 | 59.31 ±0.97 | 42.64 ±0.75 | 38.22 ±1.22 | 65.51 ±0.81 | 64.95 ±0.53 |
| | GraphSMOTE | 62.66 ±0.83 | 61.76 ±0.96 | 34.26 ±0.89 | 28.31 ±1.48 | 68.94 ±0.89 | 64.17 ±1.43 |
| | BalancedSoftmax | 67.75 ±1.09 | 66.47 ±1.27 | 52.09 ±1.61 | 48.55 ±2.01 | 71.78 ±0.82 | 71.18 ±0.94 |
| | GraphENS + TAM | 69.42 ±0.90 | 68.70 ±1.08 | 53.10 ±1.82 | 50.67 ±2.28 | 71.70 ±0.95 | 69.01 ±1.72 |
| | GraphSHA | 39.82 ±0.85 | 34.88 ±0.92 | 33.20 ±1.10 | 24.74 ±1.25 | 66.55 ±0.70 | 66.74 ±0.68 |
| | BAT1 | 72.32 ±0.52 | 71.10 ±0.55 | 59.23 ±0.90 | 58.48 ±0.87 | 73.37 ±0.45 | 72.60 ±0.48 |
| | Unreal | **78.33** ±1.04 | **76.44** ±1.06 | 65.63 ±1.38 | 64.94 ±1.38 | 75.35 ±1.41 | 73.65 ±1.43 |
| | UPL | 76.16 ±0.76 | 74.53 ±0.85 | **68.18** ±0.27 | **67.78** ±0.25 | **77.50** ±0.44 | **77.19** ±0.43 |
| GAT | Vanilla | 49.47 ±0.74 | 45.14 ±1.04 | 33.67 ±0.27 | 21.70 ±0.25 | 59.98 ±0.33 | 47.05 ±0.81 |
| | Re-Weight | 66.72 ±0.80 | 66.52 ±1.06 | 45.59 ±1.73 | 39.43 ±2.03 | 69.13 ±1.25 | 64.81 ±1.70 |
| | PC Softmax | 67.02 ±0.65 | 66.57 ±0.89 | 50.70 ±1.73 | 47.14 ±1.85 | 72.20 ±0.49 | 70.95 ±0.82 |
| | DR-GCN | 59.30 ±0.76 | 57.79 ±1.03 | 44.04 ±1.26 | 39.44 ±1.76 | 69.56 ±1.01 | 68.49 ±0.71 |
| | GraphSMOTE | 56.50 ±0.71 | 54.27 ±1.09 | 44.94 ±1.36 | 41.63 ±1.78 | 62.86 ±0.53 | 53.00 ±1.17 |
| | BalancedSoftmax | 66.63 ±0.91 | 65.66 ±0.98 | 52.51 ±1.28 | 49.71 ±1.80 | 71.09 ±0.74 | 68.95 ±0.97 |
| | GraphENS + TAM | 69.35 ±0.85 | 68.20 ±0.94 | 53.50 ±1.21 | 51.15 ±1.58 | 71.27 ±0.85 | 68.43 ±1.65 |
| | GraphSHA | 39.39 ±0.75 | 34.97 ±0.80 | 27.13 ±1.15 | 19.61 ±1.20 | 66.11 ±0.65 | 65.91 ±0.68 |
| | BAT1 | 71.4 ±0.50 | 70.2 ±0.53 | 60.6 ±0.75 | 59.9 ±0.79 | 74.6 ±0.35 | 73.9 ±0.38 |
| | Unreal | **78.91** ±0.59 | **75.99** ±0.47 | 64.10 ±1.49 | 63.44 ±1.47 | 74.68 ±1.43 | 72.78 ±0.89 |
| | UPL | 76.28 ±0.74 | 75.55 ±0.79 | **66.44** ±0.59 | **65.98** ±0.57 | **75.27** ±0.42 | **74.43** ±0.61 |
| Sage | Vanilla | 50.82 ±0.47 | 43.77 ±0.75 | 34.95 ±0.24 | 22.54 ±0.24 | 61.63 ±0.63 | 49.58 ±1.24 |
| | Re-Weight | 63.76 ±0.98 | 63.46 ±1.22 | 46.64 ±1.92 | 41.38 ±2.76 | 69.03 ±1.17 | 64.01 ±2.18 |
| | PC Softmax | 64.03 ±0.81 | 63.73 ±0.99 | 50.14 ±1.89 | 47.38 ±2.13 | 71.39 ±0.84 | 70.25 ±1.02 |
| | DR-GCN | 61.05 ±1.17 | 60.17 ±1.23 | 46.00 ±0.93 | 47.73 ±1.12 | 69.23 ±0.68 | 67.35 ±0.90 |
| | GraphSMOTE | 65.37 ±0.55 | 65.13 ±0.68 | 38.94 ±1.05 | 33.60 ±1.68 | 64.15 ±0.55 | 54.00 ±1.14 |
| | BalancedSoftmax | 63.91 ±0.88 | 63.01 ±1.00 | 49.45 ±1.65 | 46.08 ±2.08 | 70.20 ±0.72 | 69.60 ±0.86 |
| | GraphENS + TAM | 65.36 ±0.96 | 64.63 ±1.16 | 50.37 ±1.62 | 47.41 ±1.99 | 70.48 ±0.87 | 68.65 ±1.61 |
| | GraphSHA | 38.11 ±0.78 | 34.76 ±0.82 | 34.10 ±1.10 | 25.85 ±1.25 | 65.81 ±0.60 | 66.61 ±0.63 |
| | BAT1 | 71.4 ±0.51 | 70.1 ±0.54 | 60.0 ±0.78 | 59.3 ±0.81 | **76.3** ±0.33 | **75.8** ±0.36 |
| | Unreal | **75.99** ±0.98 | 73.63 ±1.23 | 66.45 ±0.39 | 65.83 ±0.30 | 74.78 ±1.30 | 72.80 ±0.54 |
| | UPL | 75.06 ±0.84 | **73.93** ±0.86 | **67.41** ±0.51 | **67.06** ±0.51 | 74.84 ±0.42 | 74.26 ±0.54 |
| GIN | Vanilla | 58.7 ±1.03 | 55.77 ±1.46 | 36.3 ±0.86 | 26.71 ±1.45 | 66.16 ±0.78 | 58.10 ±1.60 |
| | BalancedSoftmax | 66.78 ±0.79 | 65.67 ±1.02 | 46.02 ±2.19 | 41.76 ±2.77 | 70.22 ±0.80 | 66.95 ±0.92 |
| | UPL | **72.72** ±0.60 | **71.81** ±0.76 | **58.6** ±1.39 | **58.36** ±1.39 | **72.89** ±1.00 | **72.59** ±0.92 |

Table 12: Experimental results of our UPL algorithm and other baselines on three class-imbalanced node classification benchmark datasets: Chameleon, Squirrel and Wisconsin. We report averaged balanced accuracy (bAcc.) and F1-score with the standard errors for 10 repetitions on three representative GNN architectures, i.e., GCN, GAT, Sage and GIN

| | Dataset | Chameleon | | Squirrel | | Wisconsin ($\rho = 11.63$) | |
|---|---|---|---|---|---|---|---|
| | Imbalance Ratio ($\rho = 10$) | bAcc. | F1 | bAcc. | F1 | bAcc. | F1 |
| GCN | Vanilla | 57.01 ±0.53 | 55.61 ±0.67 | 38.07 ±0.57 | 34.81 ±0.69 | 34.51 ±2.31 | 31.45 ±2.66 |
| | Re-Weight | 36.07 ±0.87 | 35.61 ±0.81 | 26.92 ±0.53 | 25.04 ±0.59 | 44.13 ±3.08 | 40.74 ±3.27 |
| | PC Softmax | 36.86 ±1.04 | 36.24 ±1.01 | 26.49 ±0.59 | 25.73 ±0.49 | 30.90 ±3.10 | 28.15 ±2.16 |
| | DR-GCN | 33.34 ±0.81 | 29.60 ±0.79 | 23.34 ±0.43 | 18.20 ±0.49 | 29.44 ±1.36 | 27.08 ±1.37 |
| | GraphSMOTE | 52.38 ±0.77 | 50.03 ±0.76 | 41.30 ±0.47 | 39.07 ±0.60 | 45.36 ±4.21 | 40.91 ±4.39 |
| | BalancedSoftmax | 58.60 ±1.06 | 58.20 ±1.00 | 41.81 ±0.64 | 41.27 ±0.71 | 44.44 ±3.46 | 41.29 ±2.85 |
| | GraphENS + TAM | 40.98 ±0.85 | 38.93 ±0.84 | 27.98 ±0.90 | 24.04 ±0.70 | 41.38 ±2.78 | 34.56 ±2.22 |
| | Unreal | 58.18 ±0.94 | 56.98 ±1.03 | 45.65 ±1.08 | 44.78 ±1.07 | 41.26 ±5.19 | 32.14 ±2.15 |
| | UPL | 61.19 ±0.66 | 60.71 ±0.71 | 43.34 ±0.25 | 42.82 ±0.07 | 48.70 ±2.88 | 42.72 ±2.38 |
| GAT | Vanilla | 48.17 ±0.91 | 44.01 ±1.48 | 29.12 ±0.68 | 22.05 ±0.58 | 33.25 ±1.83 | 31.50 ±2.29 |
| | Re-Weight | 35.72 ±0.65 | 34.19 ±0.74 | 25.79 ±0.52 | 24.32 ±0.62 | 42.15 ±2.33 | 37.66 ±2.27 |
| | PC Softmax | 38.32 ±0.88 | 37.46 ±0.84 | 26.52 ±0.31 | 25.71 ±0.44 | 41.89 ±3.95 | 38.03 ±3.35 |
| | DR-GCN | 34.84 ±0.72 | 31.53 ±0.86 | 24.69 ±0.46 | 21.81 ±0.42 | 33.93 ±2.34 | 31.75 ±2.50 |
| | GraphSMOTE | 40.72 ±0.83 | 33.44 ±1.28 | 27.10 ±0.49 | 26.63 ±0.63 | 40.77 ±2.24 | 38.96 ±2.48 |
| | BalancedSoftmax | 54.37 ±0.87 | 53.91 ±0.89 | 33.73 ±1.11 | 32.82 ±1.12 | 41.05 ±2.87 | 38.65 ±2.48 |
| | GraphENS + TAM | 49.13 ±0.73 | 48.61 ±0.71 | 29.01 ±0.41 | 26.99 ±0.67 | 39.34 ±3.36 | 35.11 ±2.64 |
| | Unreal | 58.79 ±0.54 | 57.91 ±0.44 | 43.02 ±1.18 | 41.57 ±1.15 | 38.33 ±5.25 | 31.61 ±2.92 |
| | UPL | 58.87 ±0.48 | 58.72 ±0.44 | 43.62 ±0.47 | 43.07 ±0.48 | 48.35 ±3.16 | 43.34 ±2.49 |
| Sage | Vanilla | 45.61 ±0.41 | 40.52 ±0.71 | 27.55 ±0.41 | 21.43 ±0.29 | 50.08 ±3.41 | 49.39 ±4.14 |
| | Re-Weight | 36.49 ±1.21 | 34.84 ±1.30 | 29.83 ±0.59 | 25.88 ±0.42 | 68.13 ±3.19 | 63.45 ±2.27 |
| | PC Softmax | 40.71 ±0.82 | 39.95 ±0.98 | 29.23 ±0.50 | 28.19 ±0.54 | 70.57 ±3.34 | 67.13 ±2.91 |
| | DR-GCN | 39.58 ±0.58 | 38.37 ±0.72 | 28.78 ±0.50 | 25.01 ±0.70 | 69.30 ±1.99 | 64.60 ±2.00 |
| | GraphSMOTE | 33.31 ±0.63 | 30.83 ±0.67 | 25.51 ±0.43 | 19.79 ±0.49 | 65.14 ±3.84 | 62.53 ±3.40 |
| | BalancedSoftmax | 54.01 ±0.54 | 53.72 ±0.52 | 32.70 ±0.50 | 32.37 ±0.51 | 63.51 ±3.67 | 62.50 ±2.97 |
| | GraphENS + TAM | 48.01 ±0.97 | 47.19 ±1.05 | 30.75 ±0.51 | 30.54 ±0.53 | 59.69 ±3.04 | 53.28 ±2.61 |
| | Unreal | 50.21 ±1.63 | 49.35 ±1.84 | 33.43 ±0.53 | 32.30 ±0.49 | 62.20 ±6.83 | 51.52 ±4.14 |
| | UPL | 54.68 ±0.63 | 54.22 ±0.67 | 33.19 ±0.75 | 32.74 ±0.75 | 70.19 ±4.50 | 67.09 ±3.78 |
| GIN | Vanilla | 53.21 ±0.66 | 50.65 ±0.87 | 25.07 ±1.57 | 17.65 ±1.64 | 33.42 ±2.27 | 30.10 ±1.79 |
| | BalancedSoftmax | 57.43 ±1.04 | 56.81 ±1.02 | 24.24 ±1.40 | 17.10 ±1.56 | 41.12 ±2.28 | 36.54 ±1.77 |
| | UPL | 59.16 ±0.95 | 58.73 ±0.96 | 39.17 ±0.84 | 38.18 ±0.78 | 39.25 ±2.44 | 35.68 ±1.63 |

Table 13: Experimental results of our algorithm (UPL) compared to baselines for $\rho = 5$. We report the averaged balanced accuracy, averaged F1-score and the standard error of each experiment for Cora, CiteSeer and PubMed datasets. All the results have been calculated over 10 repetitions and 10 pseudo-labeling iterations.

| | Datasets | Cora | | CiteSeer | | PubMed | |
|---|---|---|---|---|---|---|---|
| | Imbalance Ratio ($\rho = 5$) | bAcc. | F1 | bAcc. | F1 | bAcc. | F1 |
| GCN | Vanilla | 63.07 ±0.76 | 62.12 ±0.98 | 39.11 ±1.21 | 30.01 ±1.89 | 67.37 ±0.96 | 59.63 ±1.76 |
| | BalancedSoftmax | 73.31 ±0.76 | 72.77 ±0.68 | 58.82 ±1.27 | 57.09 ±1.64 | 74.29 ±0.97 | 73.60 ±0.91 |
| | GraphENS + TAM | 75.25 ±0.82 | 75.00 ±0.61 | 60.68 ±1.05 | 59.86 ±1.17 | 74.18 ±0.63 | 73.21 ±0.68 |
| | UPL | 78.66 ±0.52 | 77.62 ±0.55 | 68.29 ±0.13 | 67.92 ±0.15 | 77.81 ±0.38 | 77.49 ±0.4 |
| GAT | Vanilla | 57.19 ±0.65 | 56.18 ±0.83 | 36.17 ±0.59 | 26.86 ±1.12 | 65.18 ±1.12 | 56.75 ±2.43 |
| | BalancedSoftmax | 72.48 ±0.71 | 72.05 ±0.57 | 57.74 ±0.90 | 56.17 ±1.11 | 73.77 ±0.43 | 72.85 ±0.80 |
| | GraphENS + TAM | 74.27 ±0.59 | 73.19 ±0.41 | 58.98 ±1.11 | 58.05 ±1.41 | 73.67 ±0.60 | 72.08 ±0.88 |
| | UPL | 78.94 ±0.81 | 77.8 ±0.73 | 66.92 ±0.26 | 66.44 ±0.28 | 75.83 ±0.45 | 75.63 ±0.48 |
| Sage | Vanilla | 59.30 ±0.92 | 57.30 ±1.41 | 37.21 ±0.92 | 26.80 ±1.56 | 65.88 ±0.85 | 57.60 ±1.52 |
| | BalancedSoftmax | 70.41 ±0.60 | 69.84 ±0.52 | 56.53 ±1.19 | 54.88 ±1.48 | 72.89 ±0.39 | 71.69 ±0.62 |
| | GraphENS + TAM | 72.93 ±0.35 | 72.43 ±0.41 | 59.08 ±1.19 | 58.39 ±1.27 | 74.05 ±0.79 | 73.01 ±0.97 |
| | UPL | 77.54 ±0.56 | 76.37 ±0.56 | 67.23 ±0.45 | 66.81 ±0.41 | 75.96 ±0.58 | 75.42 ±0.54 |

Table 14: Experimental results of our algorithm (UPL) compared to baselines for $\rho = 5$. We report the averaged balanced accuracy, averaged F1-score and the standard error of each experiment for Chameleon, Squirrel and Wisconsin datasets. All the results have been calculated over 10 repetitions and 10 pseudo-labeling iterations.

| | Datasets | Chameleon | | Squirrel | | Wisconsin | |
|---|---|---|---|---|---|---|---|
| | **Imbalance Ratio ($\rho = 5$)** | bAcc. | F1 | bAcc. | F1 | bAcc. | F1 |
| GCN | Vanilla | 62.03 ±0.62 | 61.62 ±0.65 | 41.48 ±0.85 | 40.38 ±0.85 | 41.03 ±2.00 | 38.97 ±1.91 |
| | BalancedSoftmax | 63.23 ±0.64 | 62.94 ±0.64 | 45.36 ±0.71 | 45.04 ±0.74 | 43.20 ±2.90 | 38.81 ±2.19 |
| | GraphENS + TAM | 43.42 ±1.05 | 41.57 ±0.95 | 27.78 ±0.46 | 23.29 ±0.74 | 40.85 ±2.93 | 36.32 ±1.78 |
| | UPL | **63.7** ±0.74 | **63.48** ±0.71 | **45.49** ±0.48 | **45.16** ±0.48 | **46.51** ±2.63 | **41.54** ±2.27 |
| GAT | Vanilla | 56.11 ±0.91 | 55.45 ±0.87 | 29.93 ±0.49 | 22.58 ±0.42 | 37.69 ±2.71 | 34.61 ±2.26 |
| | BalancedSoftmax | 57.55 ±0.96 | 57.50 ±0.85 | 39.08 ±1.38 | 38.58 ±1.43 | 39.82 ±2.30 | 37.01 ±3.14 |
| | GraphENS + TAM | 50.02 ±0.50 | 49.68 ±0.57 | 29.27 ±0.40 | 26.43 ±0.88 | **45.10** ±2.30 | 37.04 ±1.47 |
| | UPL | **61.78** ±0.8 | **61.48** ±0.8 | **46.35** ±0.58 | **46.07** ±0.57 | 44.92 ±3.18 | **41.95** ±2.84 |
| Sage | Vanilla | 49.86 ±0.40 | 48.12 ±0.61 | 29.41 ±0.51 | 25.90 ±0.58 | 61.13 ±2.03 | 62.47 ±2.21 |
| | BalancedSoftmax | 56.57 ±0.49 | 56.10 ±0.53 | 34.27 ±0.58 | 34.04 ±0.57 | 68.17 ±3.38 | 66.74 ±3.50 |
| | GraphENS + TAM | 49.14 ±0.69 | 48.42 ±0.74 | 31.90 ±0.80 | 31.39 ±0.83 | 56.33 ±4.20 | 50.60 ±3.27 |
| | UPL | **57.42** ±0.60 | **56.94** ±0.60 | **34.49** ±0.49 | **34.33** ±0.51 | **73.61** ±2.99 | **71.3** ±2.89 |

Table 15: Experimental results of our algorithm (UPL) compared to baselines for $\rho = 2$. We report the averaged balanced accuracy, averaged F1-score and the standard error of each experiment for Cora, CiteSeer and PubMed datasets. All the results have been calculated over 10 repetitions and 10 pseudo-labeling iterations.

| | Datasets | Cora | | CiteSeer | | PubMed | |
|---|---|---|---|---|---|---|---|
| | **Imbalance Ratio ($\rho = 2$)** | bAcc. | F1 | bAcc. | F1 | bAcc. | F1 |
| GCN | Vanilla | 77.09 ±0.43 | 77.18 ±0.39 | 63.29 ±0.93 | 62.74 ±0.98 | 76.93 ±0.33 | 75.70 ±0.40 |
| | BalancedSoftmax | 79.46 ±0.28 | 78.58 ±0.30 | 67.26 ±0.49 | 66.84 ±0.53 | 78.25 ±0.28 | 77.78 ±0.29 |
| | GraphENS + TAM | 79.88 ±0.32 | 79.15 ±0.35 | 67.27 ±0.34 | 66.87 ±0.36 | 76.42 ±0.48 | 76.15 ±0.51 |
| | UPL | **81.24** ±0.31 | **80.27** ±0.30 | **68.58** ±0.29 | **68.16** ±0.31 | **78.37** ±0.21 | **77.88** ±0.21 |
| GAT | Vanilla | 73.61 ±0.49 | 74.45 ±0.49 | 60.76 ±1.19 | 59.84 ±1.49 | 74.43 ±0.55 | 72.14 ±0.81 |
| | BalancedSoftmax | 78.68 ±0.28 | 77.64 ±0.35 | 65.71 ±0.39 | 65.14 ±0.40 | 76.54 ±0.37 | 75.49 ±0.49 |
| | GraphENS + TAM | 79.25 ±0.27 | 78.13 ±0.28 | 66.62 ±0.40 | 66.24 ±0.40 | 76.28 ±0.47 | 75.87 ±0.46 |
| | UPL | **80.53** ±0.32 | **79.76** ±0.19 | **67.69** ±0.23 | **67.31** ±0.21 | **77.83** ±0.22 | **77.04** ±0.29 |
| Sage | Vanilla | 74.44 ±0.25 | 74.63 ±0.24 | 62.32 ±0.70 | 61.63 ±0.80 | 75.65 ±0.68 | 74.41 ±0.76 |
| | BalancedSoftmax | 77.56 ±0.27 | 76.61 ±0.25 | 65.74 ±0.48 | 65.24 ±0.49 | 75.68 ±0.40 | 74.97 ±0.38 |
| | GraphENS + TAM | 78.79 ±0.41 | 77.07 ±0.42 | 66.83 ±0.27 | 66.32 ±0.30 | 76.28 ±0.46 | 75.71 ±0.47 |
| | UPL | **80.12** ±0.26 | **78.67** ±0.31 | **68.98** ±0.20 | **68.62** ±0.19 | **76.90** ±0.30 | **76.45** ±0.26 |

Table 16: Experimental results of our algorithm (UPL) compared to baselines for $\rho = 2$. We report the averaged balanced accuracy, averaged F1-score and the standard error of each experiment for Chameleon, Squirrel and Wisconsin datasets. Same as other experiments, all the results have been calculated over 10 repetitions and 10 pseudo-labeling iterations.

| | Datasets | Chameleon | | Squirrel | | Wisconsin | |
|---|---|---|---|---|---|---|---|
| | **Imbalance Ratio ($\rho = 2$)** | bAcc. | F1 | bAcc. | F1 | bAcc. | F1 |
| GCN | Vanilla | 66.95 ±0.73 | 66.67 ±0.69 | 49.53 ±0.38 | 49.59 ±0.38 | 39.63 ±2.14 | 38.28 ±2.26 |
| | BalancedSoftmax | 67.10 ±0.79 | 66.78 ±0.73 | 50.00 ±0.47 | 49.74 ±0.48 | **44.81** ±3.18 | **41.32** ±2.32 |
| | Graphens + TAM | 43.76 ±1.08 | 41.49 ±1.38 | 26.98 ±0.79 | 23.69 ±0.67 | 43.86 ±2.89 | 37.68 ±2.15 |
| | UPL | **67.58** ±0.56 | **67.37** ±0.54 | **51.26** ±0.62 | **51.01** ±0.56 | 44.32 ±3.27 | 40.30 ±2.41 |
| GAT | Vanilla | 63.44 ±0.50 | 63.06 ±0.45 | 39.20 ±1.24 | 38.06 ±1.55 | 34.40 ±2.06 | 32.28 ±1.72 |
| | BalancedSoftmax | 62.81 ±0.93 | 62.39 ±0.89 | 43.99 ±1.95 | 43.55 ±1.98 | 42.17 ±2.41 | 37.66 ±2.74 |
| | Graphens + TAM | 53.21 ±0.84 | 52.86 ±0.93 | 33.21 ±0.58 | 32.81 ±0.65 | 44.02 ±2.16 | 36.57 ±1.60 |
| | UPL | **65.68** ±0.57 | **65.39** ±0.55 | **50.13** ±0.39 | **49.68** ±0.35 | **44.40** ±2.51 | **41.34** ±2.44 |
| Sage | Vanilla | 59.18 ±0.42 | 58.85 ±0.41 | 36.10 ±0.63 | 35.98 ±0.63 | 65.57 ±2.81 | 66.42 ±2.48 |
| | BalancedSoftmax | 59.17 ±0.52 | 58.55 ±0.50 | 37.22 ±0.53 | **37.00** ±0.52 | 74.10 ±2.77 | 70.37 ±3.04 |
| | Graphens + TAM | 51.25 ±0.90 | 50.87 ±0.93 | 32.86 ±0.49 | 32.52 ±0.51 | 63.35 ±3.53 | 53.89 ±3.07 |
| | UPL | **60.01** ±0.56 | **59.59** ±0.57 | **37.30** ±0.58 | 36.94 ±0.58 | **75.21** ±3.21 | **73.20** ±2.86 |

Table 17: Experimental results of our algorithm (UPL) compared to baselines for $\rho = 20$. We report the averaged balanced accuracy, averaged F1-score and the standard error of each experiment for Cora, CiteSeer and PubMed datasets. All the results have been calculated over 10 repetitions and 10 pseudo-labeling iterations.

| Datasets | | Cora | | CiteSeer | | PubMed | |
|---|---|---|---|---|---|---|---|
| **Imbalance Ratio ($\rho = 20$)** | | bAcc. | F1 | bAcc. | F1 | bAcc. | F1 |
| GCN | Vanilla | 50.01 ±0.18 | 42.37 ±0.28 | 34.95 ±0.23 | 21.83 ±0.11 | 61.11 ±0.19 | 46.91 ±0.43 |
| | BalancedSoftmax | 63.75 ±1.11 | 62.67 ±1.18 | 47.83 ±1.67 | 42.52 ±1.88 | 69.52 ±1.03 | 68.00 ±0.89 |
| | Graphens + Tam | 64.29 ±1.42 | 62.85 ±1.55 | 48.66 ±1.78 | 45.03 ±2.19 | 66.65 ±1.26 | 60.53 ±2.42 |
| | UPL | **73.92** ±0.90 | **73.48** ±0.98 | **64.97** ±1.18 | **64.18** ±1.29 | **76.67** ±0.20 | **76.61** ±0.20 |
| GAT | Vanilla | 48.27 ±0.26 | 41.45 ±0.21 | 34.28 ±0.22 | 21.80 ±0.18 | 59.43 ±0.27 | 45.36 ±0.32 |
| | BalancedSoftmax | 61.22 ±1.07 | 60.64 ±1.08 | 47.21 ±0.95 | 43.03 ±1.16 | 68.55 ±0.69 | 65.84 ±0.91 |
| | Graphens + Tam | 64.28 ±1.47 | 62.80 ±1.70 | 48.71 ±1.48 | 44.67 ±1.76 | 67.15 ±1.66 | 60.29 ±2.95 |
| | UPL | **72.15** ±0.74 | **71.48** ±0.73 | **60.75** ±1.46 | **60.25** ±1.47 | **75.39** ±0.42 | **75.13** ±0.38 |
| Sage | Vanilla | 49.50 ±0.17 | 41.93 ±0.32 | 34.65 ±0.09 | 22.05 ±0.08 | 60.15 ±0.36 | 46.89 ±0.75 |
| | BalancedSoftmax | 60.55 ±1.14 | 59.39 ±1.40 | 44.23 ±1.21 | 38.60 ±1.37 | 67.10 ±0.44 | 66.58 ±0.55 |
| | Graphens + Tam | 59.39 ±1.01 | 57.18 ±1.25 | 42.26 ±1.54 | 35.89 ±1.91 | 66.03 ±1.01 | 59.67 ±2.06 |
| | UPL | **72.45** ±1.02 | **71.80** ±1.13 | **65.39** ±0.75 | **64.89** ±0.90 | **73.09** ±0.55 | **73.19** ±0.49 |
| GIN | Vanilla | 51.24 ±0.95 | 45.53 ±1.39 | 34.16 ±0.23 | 22.49 ±0.36 | 61.8 ±0.63 | 49.47 ±1.44 |
| | BalancedSoftmax | 61.23 ±0.79 | 59.58 ±1.08 | 42.12 ±1.39 | 36.85 ±1.75 | 65.24 ±1.37 | 60.24 ±1.51 |
| | UPL | **68.92** ±0.65 | **67.94** ±0.70 | **54.06** ±1.35 | **53.65** ±1.41 | **75.92** ±0.37 | **75.23** ±0.36 |

Table 18: Experimental results of our algorithm (UPL) compared to baselines for $\rho = 20$. We report the averaged balanced accuracy, averaged F1-score and the standard error of each experiment for Chameleon, Squirrel and Wisconsin datasets. All the results have been calculated over 10 repetitions and 10 pseudo-labeling iterations.

| Datasets | | Chameleon | | Squirrel | | Wisconsin | |
|---|---|---|---|---|---|---|---|
| **Imbalance Ratio ($\rho = 20$)** | | bAcc. | F1 | bAcc. | F1 | bAcc. | F1 |
| GCN | Vanilla | 52.14 ±0.62 | 48.43 ±0.97 | 36.25 ±0.48 | 31.02 ±0.63 | 32.28 ±1.24 | 28.84 ±1.54 |
| | BalancedSoftmax | 54.63 ±0.79 | 53.10 ±0.70 | **39.66** ±0.53 | **38.51** ±0.55 | 40.18 ±3.31 | 36.53 ±2.35 |
| | Graphens + Tam | 53.27 ±0.74 | 52.34 ±0.80 | 38.87 ±0.41 | 38.01 ±0.46 | **44.43** ±4.13 | 37.20 ±2.01 |
| | UPL | **55.98** ±0.80 | **54.99** ±0.83 | 39.01 ±0.60 | 38.10 ±0.60 | 42.61 ±3.35 | **38.16** ±2.78 |
| GAT | Vanilla | 43.82 ±0.54 | 36.15 ±0.91 | 29.84 ±0.76 | 22.42 ±0.63 | 30.98 ±0.99 | 27.92 ±1.15 |
| | BalancedSoftmax | 51.26 ±0.55 | 50.94 ±0.65 | 31.72 ±1.08 | 30.80 ±1.08 | 38.63 ±3.28 | 35.45 ±2.19 |
| | Graphens + Tam | 47.19 ±0.82 | 46.19 ±0.92 | 28.80 ±0.60 | 26.86 ±0.99 | 43.27 ±4.25 | 34.73 ±2.05 |
| | UPL | **54.62** ±0.67 | **54.07** ±0.71 | **40.83** ±0.51 | **39.68** ±0.50 | **43.60** ±3.14 | **41.69** ±2.68 |
| Sage | Vanilla | 43.98 ±0.58 | 36.68 ±1.12 | 27.33 ±0.46 | 20.69 ±0.37 | 42.45 ±2.71 | 39.62 ±2.99 |
| | BalancedSoftmax | 49.21 ±0.97 | 48.94 ±0.83 | 32.17 ±0.55 | 31.63 ±0.58 | 54.16 ±4.31 | 52.16 ±4.16 |
| | Graphens + Tam | 44.62 ±0.98 | 43.60 ±0.86 | 30.04 ±0.45 | 29.31 ±0.39 | 48.54 ±4.17 | 44.28 ±2.58 |
| | UPL | **50.63** ±0.63 | **50.58** ±0.53 | **32.50** ±0.52 | **32.08** ±0.54 | **55.47** ±2.89 | **52.76** ±2.32 |
| GIN | Vanilla | 49.15 ±0.64 | 44.38 ±0.83 | 25.23 ±1.52 | 18.03 ±1.38 | 32.47 ±0.66 | 30.35 ±0.78 |
| | BalancedSoftmax | 52.39 ±1.05 | 51.17 ±1.13 | 23.7 ±1.24 | 18.32 ±1.81 | **41.17** ±2.27 | **37.95** ±1.60 |
| | UPL | **54.41** ±1.05 | **53.52** ±1.05 | **31.69** ±1.02 | **30.08** ±1.08 | 39.27 ±2.25 | 35.85 ±1.31 |

## H.2  Additional Datasets

To further test UPL, we have included additional experiments with Amazon datasets (Shchur et al., 2019), comparing them to other previous works in Table 19, Table 20, and ablation studies in Table 21.

Table 19: Experimental results of our algorithm (UPL) compared to baselines for $\rho = 10$. We report the averaged balanced accuracy, averaged F1-score and the standard error of each experiment for Computers and Photo datasets. All the results have been calculated over 10 repetitions and 10 pseudo-labeling iterations.

| | Datasets | Computers | | Photo | |
|---|---|---|---|---|---|
| | **Imbalance Ratio ($\rho = 10$)** | bAcc. | F1 | bAcc. | F1 |
| GCN | Vanilla | 47.11 $\pm2.28$ | 36.53 $\pm3.27$ | 53.80 $\pm3.09$ | 44.83 $\pm4.50$ |
| | BalancedSoftmax | 74.35 $\pm1.30$ | 73.73 $\pm1.35$ | 74.86 $\pm1.20$ | 74.33 $\pm1.39$ |
| | GraphENS + TAM | 73.82 $\pm2.76$ | 72.68 $\pm3.02$ | **82.34** $\pm1.76$ | **81.39** $\pm2.26$ |
| | UPL | **75.53** $\pm0.94$ | **75.84** $\pm0.97$ | 79.27 $\pm0.93$ | 78.67 $\pm0.93$ |
| GAT | Vanilla | 41.24 $\pm1.24$ | 32.45 $\pm1.92$ | 46.52 $\pm0.67$ | 37.62 $\pm1.55$ |
| | BalancedSoftmax | 64.11 $\pm2.09$ | 63.01 $\pm2.70$ | 71.36 $\pm1.90$ | 69.78 $\pm2.07$ |
| | GraphENS + TAM | 64.55 $\pm0.95$ | 62.59 $\pm1.34$ | 68.86 $\pm1.34$ | 65.52 $\pm1.60$ |
| | UPL | **74.62** $\pm1.08$ | **74.20** $\pm1.10$ | **81.93** $\pm0.87$ | **81.60** $\pm0.91$ |
| Sage | Vanilla | 48.16 $\pm1.84$ | 38.56 $\pm2.70$ | 60.43 $\pm2.83$ | 52.07 $\pm3.69$ |
| | BalancedSoftmax | 73.18 $\pm1.51$ | 72.19 $\pm1.71$ | 78.55 $\pm1.41$ | 77.46 $\pm1.59$ |
| | GraphENS + TAM | 75.20 $\pm1.47$ | 73.96 $\pm1.72$ | 81.36 $\pm1.54$ | 80.67 $\pm1.64$ |
| | UPL | **81.56** $\pm0.94$ | **81.49** $\pm1.01$ | **89.00** $\pm0.44$ | **88.90** $\pm0.46$ |
| GIN | Vanilla | 51.76 $\pm6.22$ | 47.36 $\pm6.82$ | 74.3 $\pm1.20$ | 74.25 $\pm1.37$ |
| | BalancedSoftmax | 47.27 $\pm7.61$ | 43.37 $\pm8.70$ | 79.66 $\pm1.02$ | 79.43 $\pm1.08$ |
| | UPL | **75.84** $\pm1.19$ | **75.61** $\pm1.34$ | **82.7** $\pm0.71$ | **82.66** $\pm0.71$ |

Table 20: Experimental results of our algorithm (UPL) compared to baselines for $\rho = 20$. We report the averaged balanced accuracy, averaged F1-score and the standard error of each experiment for Computers and Photo datasets. All the results have been calculated over 10 repetitions and 10 pseudo-labeling iterations.

| | Datasets | Computers | | Photo | |
|---|---|---|---|---|---|
| | **Imbalance Ratio ($\rho = 20$)** | bAcc. | F1 | bAcc. | F1 |
| GCN | Vanilla | 42.85 $\pm0.37$ | 30.79 $\pm0.33$ | 45.82 $\pm0.23$ | 33.49 $\pm0.21$ |
| | BalancedSoftmax | 64.33 $\pm1.63$ | 63.93 $\pm1.78$ | 66.27 $\pm2.54$ | 64.84 $\pm2.68$ |
| | Graphens + Tam | **73.38** $\pm1.72$ | **71.33** $\pm2.06$ | **78.50** $\pm1.48$ | **77.24** $\pm1.78$ |
| | UPL | 66.29 $\pm1.65$ | 65.65 $\pm1.93$ | 71.86 $\pm1.87$ | 70.60 $\pm1.97$ |
| GAT | Vanilla | 40.95 $\pm0.78$ | 31.45 $\pm0.48$ | 44.82 $\pm0.62$ | 34.36 $\pm0.47$ |
| | BalancedSoftmax | 59.27 $\pm1.71$ | 57.27 $\pm2.29$ | 60.73 $\pm1.62$ | 58.45 $\pm1.83$ |
| | Graphens + Tam | 58.76 $\pm1.54$ | 57.52 $\pm1.71$ | 67.59 $\pm1.84$ | 65.23 $\pm2.04$ |
| | UPL | **66.67** $\pm1.31$ | **65.78** $\pm1.36$ | **76.41** $\pm0.99$ | **75.10** $\pm1.25$ |
| Sage | Vanilla | 45.36 $\pm0.28$ | 33.79 $\pm0.20$ | 46.75 $\pm0.18$ | 34.34 $\pm0.23$ |
| | BalancedSoftmax | 67.15 $\pm1.30$ | 65.60 $\pm1.40$ | 72.39 $\pm1.62$ | 71.49 $\pm1.75$ |
| | Graphens + Tam | 67.16 $\pm2.19$ | 64.69 $\pm2.41$ | 72.05 $\pm2.73$ | 69.94 $\pm3.14$ |
| | UPL | **75.95** $\pm1.33$ | **74.94** $\pm1.57$ | **85.80** $\pm1.22$ | **85.15** $\pm1.45$ |
| GIN | Vanilla | 37.98 $\pm4.74$ | 31.59 $\pm4.83$ | 64.91 $\pm0.88$ | 62.29 $\pm1.14$ |
| | BalancedSoftmax | 45.71 $\pm6.12$ | 43.55 $\pm6.74$ | 72.82 $\pm1.60$ | 71.62 $\pm2.01$ |
| | UPL | **71.71** $\pm2.34$ | **71.04** $\pm2.58$ | **77.52** $\pm1.47$ | **77.10** $\pm1.53$ |

We have also included some experiments to compare our performance with Unreal under massive graph datasets flickr. These result are displayed in Table 22.

## H.3  Impact of Pseudo-Labeling Scope

A fundamental design choice in the UPL framework is the restriction of pseudo-labeling exclusively to minority classes. Our core hypothesis is that in imbalanced semi-supervised scenarios, the classifier is inherently biased

Table 21: We conducted an ablation study on our UPL algorithm, comparing it to baselines by systematically removing key components. We report the averaged balanced accuracy, averaged F1-score, and the standard error for each experiment on the Computers and Photo datasets. The results were calculated over 10 repetitions and 10 pseudo-labeling iterations.

| | Datasets | Computers | | Photo | |
|---|---|---|---|---|---|
| | **Imbalance Ratio ($\rho = 10$)** | bAcc. | F1 | bAcc. | F1 |
| GCN | Vanilla | 46.15 ±1.61 | 35.25 ±2.37 | 48.30 ±2.46 | 36.86 ±3.46 |
| | + BS | 74.24 ±0.92 | 74.57 ±0.92 | 76.86 ±0.87 | 75.97 ±0.92 |
| | + PL | 61.95 ±1.55 | 57.85 ±1.92 | 63.95 ±2.97 | 58.74 ±4.17 |
| | + Uncertainty | 47.96 ±2.25 | 37.78 ±3.19 | 48.25 ±2.42 | 36.79 ±3.40 |
| | + BS + PL | 73.56 ±1.17 | 73.74 ±1.14 | 76.86 ±0.99 | 75.92 ±1.08 |
| | + Uncertainty + BS | 74.44 ±1.02 | 74.54 ±1.07 | 76.61 ±0.92 | 75.91 ±0.96 |
| | + Uncertainty + PL | 62.20 ±3.10 | 55.43 ±3.83 | 72.14 ±2.93 | 68.49 ±3.95 |
| | UPL | **75.53** ±0.94 | **75.84** ±0.97 | **79.27** ±0.93 | **78.67** ±0.93 |

Table 22: Experimental results of our algorithm (UPL) compared to Unreal for the large dataset of Flickr. We report the averaged balanced accuracy, averaged F1-score and the standard error. All the results have been calculated over 10 repetitions and 10 pseudo-labeling iterations.

| Backbone | GCN | | GAT | | SAGE | |
|---|---|---|---|---|---|---|
| | bAcc. | F1 | bAcc. | F1 | bAcc. | F1 |
| Unreal | 22.82 ±0.27 | 21.31 ±0.47 | 23.82 ±1.62 | 18.64 ±1.18 | 24.57 ±0.41 | 22.88 ±0.50 |
| **UPL** | **28.57** ±0.34 | **24.07** ±0.44 | **27.32** ±0.17 | **23.46** ±0.28 | **30.60** ±0.08 | **25.97** ±0.23 |

toward majority classes. Consequently, indiscriminate pseudo-labeling (applying labels to all classes) risks amplifying this bias, as the model is likely to confidently misclassify ambiguous nodes as majority class samples.

To empirically validate this hypothesis, we evaluated a variant of our method termed **UPL-AC (UPL-All Classes)**, where the minority-only constraint is removed, allowing pseudo-labels to be assigned to any node satisfying the uncertainty thresholds. We tested this variant both with and without the CWCE (CWCE) loss function.

The results, detailed in Table 23, provide strong evidence supporting the minority-only strategy. We observe the following key trends:

1. **Degradation via Indiscriminate Labeling:** Across all backbones (GCN, GAT, SAGE, GIN), applying pseudo-labels to all classes consistently degrades performance compared to the proposed UPL. For instance, on the Cora dataset using GCN, UPL-AC results in a 4.47% drop in balanced accuracy compared to UPL.

2. **Confirmation Bias without Re-weighting:** The effect is most severe when the CWCE loss is removed (`UPL-AC w/o CWCE`). In this setting, the model's performance collapses (e.g., F1 score drops to 47.82 on Cora). This confirms that without the minority-only constraint, the self-training loop simply reinforces the majority class dominance, effectively ignoring the minority samples.

3. **Synergy of UPL components:** Even when CWCE is used to mitigate loss-level imbalance (`UPL-AC + CWCE`), the performance remains inferior to the proposed UPL. This indicates that loss re-weighting alone is insufficient to counteract the noise introduced by majority-class pseudo-labels.

These findings demonstrate that "more data" is not intrinsically better in imbalanced graph learning; rather, the distributional alignment of the pseudo-labeled set is the governing factor for success.

Table 23: This table compares the outcome of applying pseudo labeling to all classes with BS (UPL-AC + BS) and without BS (UPL-AC w/o BS) to proposed method (UPL).

| $\rho = 10$ | | UPL | | UPL-AC + BS ($\Delta$ from UPL) | | UPL-AC w/o BS ($\Delta$ from UPL-AC+BS) | |
|---|---|---|---|---|---|---|---|
| | | bAcc. | F1 | bAcc. | F1 | bAcc. | F1 |
| GCN | Cora | **76.16** | **74.53** | 71.69 (-4.47) | 71.26 (-3.27) | 53.12 (-18.57) | 47.82 (-23.44) |
| | CiteSeer | **68.18** | **67.78** | 65.38 (-2.80) | 64.02 (-3.76) | 35.37 (-30.01) | 22.76 (-41.26) |
| | PubMed | **77.50** | **77.19** | 76.07 (-1.43) | 73.58 (-3.61) | 62.08 (-13.99) | 49.04 (-24.54) |
| GAT | Cora | **76.28** | **75.55** | 71.24 (-5.04) | 70.14 (-5.41) | 49.65 (-21.59) | 45.05 (-25.09) |
| | CiteSeer | **66.44** | **65.98** | 63.03 (-3.41) | 62.16 (-3.82) | 34.15 (-28.88) | 22.88 (-39.28) |
| | PubMed | **75.27** | **74.43** | 72.71 (-2.56) | 70.77 (-3.66) | 62.03 (-10.68) | 53.03 (-17.74) |
| SAGE | Cora | **75.06** | **73.93** | 69.90 (-5.16) | 69.35 (-4.58) | 49.95 (-19.95) | 43.70 (-25.65) |
| | CiteSeer | **67.41** | **67.06** | 63.79 (-3.62) | 63.11 (-3.95) | 33.85 (-29.94) | 21.93 (-41.18) |
| | PubMed | **74.84** | **74.26** | 73.69 (-1.15) | 71.11 (-3.15) | 61.65 (-12.04) | 49.90 (-21.21) |

## H.4 Uncertainty methods

We evaluate our approach's uncertainty estimation on the Cora dataset, comparing against two uncertainty estimation methods for GNN as baselines: CaGCN Wang et al. (2021) and ALG Zhang et al. (2021b) in Table 24. Our results show that imbalanced data significantly impacts uncertainty estimation performance in existing graph-based models.

Table 24: Comparison of our model with CaGCN and Alg as uncertainty estimation in GNN utilizing uncertainty in graphs has been tested under the imbalance scenario with ratio $\rho = 10$.

| **Methods** | **Cora** | **CiteSeer** |
|---|---|---|
| CaGCN | 46.23 $\pm 0.49$ | 23.42 $\pm 0.74$ |
| Alg | 67.48 $\pm 0.76$ | 27.50 $\pm 0.85$ |
| UPL | 74.51 $\pm 0.65$ | 67.78 $\pm 0.25$ |

## H.5 Convergence and Stabilization Analysis

The UPL algorithm repeats pseudo-label assignment and retraining for $k$ iterations. To empirically evaluate convergence and determine a practical stopping point, we performed a sweep over the number of pseudo-labeling iterations ($k$) from 2 to 20, using a step size of 2.

As demonstrated in Table 25, performance largely stabilizes around $k = 10$. While minor fluctuations and marginal improvements occasionally occur after this point—such as the late jump observed in CiteSeer at iteration 18—the vast majority of the performance gains are mostly captured within the first 10 iterations. This stabilization plateau is particularly apparent in the Cora dataset. These empirical results justify our choice of $k = 10$ as it provides a robust balance between computational efficiency and predictive accuracy.

## H.6 Discussion

Both Table 11 and Table 12 demonstrate a clear leading performance of our method across various architectures and datasets. A particularly notable aspect of our experiments is the significant margin by which our results surpass the second-best outcomes. This pronounced difference underscores the effectiveness of our approach, setting a new benchmark for performance in this field.

Our experimental analysis investigated the impact of varying the number of edge removal iterations ($t$) and the number of edges removed in each iteration ($S_k$) on model performance. The results, provided in Figure

Table 25: Performance Metrics by Dataset and Iteration Step ($k$) with Step-over-Step Differences

| Dataset | Step ($k$) | bAcc. | $\Delta$ bAcc. | F1 | $\Delta$ F1 |
|---|---|---|---|---|---|
| CiteSeer | 2 | 53.77 | – | 51.63 | – |
| | 4 | 53.94 | +0.17 | 52.16 | +0.53 |
| | 6 | 55.06 | +1.12 | 53.37 | +1.21 |
| | 8 | 56.08 | +1.02 | 54.38 | +1.01 |
| | 10 | 56.19 | +0.11 | 54.68 | +0.30 |
| | 12 | 56.45 | +0.26 | 54.98 | +0.30 |
| | 14 | 56.79 | +0.34 | 55.30 | +0.32 |
| | 16 | 56.83 | +0.04 | 55.29 | -0.01 |
| | 18 | 57.84 | +1.01 | 56.44 | +1.15 |
| | 20 | 58.11 | +0.27 | 56.83 | +0.39 |
| Cora | 2 | 70.29 | – | 69.51 | – |
| | 4 | 72.01 | +1.72 | 71.12 | +1.61 |
| | 6 | 72.63 | +0.62 | 71.88 | +0.76 |
| | 8 | 73.01 | +0.38 | 72.22 | +0.34 |
| | 10 | 73.08 | +0.07 | 72.53 | +0.31 |
| | 12 | 73.70 | +0.62 | 73.20 | +0.67 |
| | 14 | 73.49 | -0.21 | 72.96 | -0.24 |
| | 16 | 73.84 | +0.35 | 73.35 | +0.39 |
| | 18 | 73.97 | +0.13 | 73.55 | +0.20 |
| | 20 | 73.69 | -0.28 | 73.25 | -0.30 |

4, demonstrate that choosing both $t = 100$ and $S_k = 100$ yields the best results in both studied datasets. Notably, the computation of uncertainty values does not require any training steps, which significantly reduces the computational overhead. Given that typical model training procedures require over 1000 epochs, the time needed to calculate uncertainty values is negligible in comparison to the overall training process.

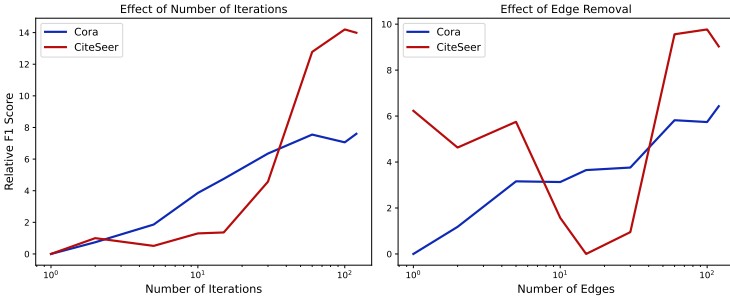

Figure 4: Selection Edge Removal: F1 score versus number of iterations and number of edges for removal. Each plot is normalized to it's lowest value.

Table 26: UPL performance summary: Comparison of UPL in terms of the number of Best|Second rank performance.

| Datasets Types | GCN | GAT | SAGE |
|---|---|---|---|
| Homophilic | 12|4 | 15|1 | 15|1 |
| Heterophilic | 9|3 | 12|0 | 9|3 |
| Total | 21|7 | 27|1 | 24|4 |

For a summary of the performance of UPL, we have provided Table 26, comparing our model with previous methods in term of F1 score ranking based on topological nature of the dataset and the employed backbone. Performance ranking is measured in terms of Best|Second. It is worth noting that this ranking is adequate as UPL hasn't performed lower than second compared to previous methods in any of the experiments.

### H.7 Ablation on Selective Edge Removal (SER)

It is important to clarify the independent role of Selective Edge Removal (SER) within our framework. As shown in Table 3, utilizing uncertainty alone ("+ Uncertainty") matches the Vanilla baseline's performance. This is an expected behavior: SER does not create a training signal by itself; rather, it acts strictly as a filter for candidate pseudo-labels. Therefore, in a context where no pseudo-labels are generated, the uncertainty filter has no labels to process, resulting in a model identical to the Vanilla baseline. To properly evaluate the value of SER, it must be compared within a pseudo-labeling context (e.g., comparing the "+ PL" configuration against the full "UPL" framework). In these scenarios, the addition of the SER filter demonstrably improves classification performance.

## I Detailed Comparison with UNREAL

**Comparison with UNREAL:** Our algorithm, UPL, demonstrates notable computational advantages over the UNREAL algorithm (Yan et al., 2023). UPL's efficiency stems from two key factors: first, it employs a more selective pseudo-labeling process that focuses on minority classes, resulting in fewer pseudo-labeled nodes overall. Second, UPL eliminates the need for two computationally intensive components present in UNREAL—node reordering and clustering—which contribute significantly to that algorithm's complexity. Instead, UPL introduces a novel framework that incorporates uncertainty estimation and an upper threshold for pseudo-labeling. In contrast, the UNREAL algorithm operates through a series of three main steps,

- Dual Pseudo-tag Alignment Mechanism (DPAM): This step involves clustering all nodes, which inherently have a complexity of $O(N^2)$ for $N$ nodes, and then inferring the GCN model.

- Node-Reordering: Utilizes the Rank-Biased Overlap (RBO) algorithm (Webber et al., 2010) to merge two sorted lists generated by clustering and GCN inference. This merging process also presents a complexity of $O(N^2)$ for $N$ nodes.

- Discarding Geometrically Imbalanced Nodes (DGIN): This step computes the second nearest cluster center for each node, leading to a complexity of $O(Nk)$, where $k$ is the predefined number of clusters,

where each is iterated $T$ times, potentially leading to increased computational overhead. By reducing both the number of operations each step, UPL achieves improved efficiency without sacrificing performance. In particular, our UPL algorithm simplifies the computational process significantly:

- Uncertainty estimation: UPL involves sampling the edges to be removed based on their distribution to estimate the uncertainty. This process is iterated $S_k$ times.

- Subsequently, similar to UNREAL, we infer the GCN model.

Notably our complexity is less than the UNREAL, as we avoid DGIN and RBO processes in UNREAL.

Furthermore, to assess practical efficiency, we conducted time measurements for both UNREAL and UPL under identical experimental conditions. Our findings demonstrate that UPL's streamlined approach, which employs fewer and less complex iterative processes, substantially reduces both time and computational resource requirements, resulting in superior empirical efficiency. This advantage is clearly illustrated in our experiments with the PubMed dataset: UPL completed 10 runs in an average runtime of 0.66 minutes, while UNREAL required an average runtime of 14.1 minutes for the same number of runs. These timing tests were performed on a server equipped with an Nvidia RTX 3090 GPU and AMD Ryzen 5 3600x CPU @ 3.80GHz, ensuring consistent and comparable results. While UPL uses less time to run, it also delivers better

performance, as shown in Tables 1, 2, 11, and 12. Our extensive evaluation across 24 different experimental setups reveals that UPL outperforms UNREAL in about 80% of the cases. Compared to all methods, in 17 out of these 24 setups, UPL achieved better results in both balanced accuracy and F1 score, while in the remaining cases, it secured second place, highlighting its robustness and consistent performance across a variety of conditions.

**Performance on Heterophilic Datasets:** While UPL delivers superior performance across most scenarios, we acknowledge that it is highly competitive but not uniformly the best on all heterophilic datasets. Specifically, on the Squirrel dataset utilizing the GCN backbone, UNREAL outperforms UPL in both balanced accuracy and F1 score. This indicates that our degree-biased SER strategy may not always perfectly align with reliability under conditions of strong heterophily and severe degree heterogeneity. Exploring heterophily-aware perturbation designs for uncertainty estimation remains a promising direction for future work. Nonetheless, it is important to highlight that across 9 distinct experimental tests in the heterophilic domain (Chameleon, Squirrel, Wisconsin across various backbones), UPL is outperformed only once, maintaining a highly robust success rate of 89%.

## J  Statistical Significance of Results

To compare the significance of our model against the BAT1 and Unreal baselines, we calculated P-values using Welch's t-test for key pairwise comparisons. The results are displayed in Table 27. It is important to note that each UPL result is derived from 10 independent experimental runs, whereas BAT1 and Unreal report results based on 5 runs. While this difference in sample sizes means P-values should be interpreted with some caution, the analysis nonetheless demonstrates statistically significant advantages ($p < 0.05$) for UPL in several key scenarios, particularly against BAT1 across multiple backbones and datasets.

Table 27: P-values comparing UPL against the Unreal and BAT1 baselines on imbalanced graph node classification datasets. P-values are calculated using Welch's t-test based on 10 runs for UPL and 5 runs for the baselines. Statistically significant differences ($p < 0.05$) are highlighted in bold.

| | **Comparison** | Cora | | CiteSeer | | PubMed | |
|---|---|---|---|---|---|---|---|
| | **Metric** | bAcc. | F1 | bAcc. | F1 | bAcc. | F1 |
| GCN | UPL vs. Unreal | 0.130 | 0.190 | 0.140 | 0.110 | 0.210 | 0.070 |
| GCN | UPL vs. BAT1 | **0.001** | **0.005** | **< 0.001** | **< 0.001** | **< 0.001** | **< 0.001** |
| GAT | UPL vs. Unreal | **0.018** | 0.640 | 0.200 | 0.170 | 0.710 | 0.160 |
| GAT | UPL vs. BAT1 | **< 0.001** | **< 0.001** | **< 0.001** | **< 0.001** | 0.240 | 0.470 |
| Sage | UPL vs. Unreal | 0.490 | 0.840 | 0.160 | 0.057 | 0.960 | 0.082 |
| Sage | UPL vs. BAT1 | **0.002** | **0.002** | **< 0.001** | **< 0.001** | **0.018** | **0.034** |

## K  Extended Theoretical Results

In this section, we provide extended theoretical results, including,

- Multi-class scenario

- Pseudo-labeling

- Uncertainty-aware selection

### K.1    Multi-class extension via pairwise margins

This section gives a multi-class extension. For this purpose, we apply the scalar transductive Rademacher bound of El-Yaniv & Pechyony (2009) to scalar pairwise margin losses and then union-bounds over ordered class pairs.

Assume that we have $K$ classes. Let the fixed transductive sample be

$$\mathcal{Z} = \{(x_i, y_i)\}_{i=1}^N, \qquad y_i \in [K].$$

For each class $c \in [K]$, let

$$I_c = \{i : y_i = c\}, \qquad |I_c| = n_c = m_c + u_c.$$

Condition on the realized class counts. Within each class $c$, assume the labeled subset $S_c \subset I_c$ of size $m_c$ is drawn uniformly without replacement, and the transductive test subset is $T_c = I_c \setminus S_c$ with $|T_c| = u_c$.

Let $h : X \to \mathbb{R}^K$ be a score function and

$$\widehat{y}_h(x) = \arg\max_{a \in [K]} h_a(x).$$

The multi-class transductive test error is

$$L_u^K(h) = \frac{1}{u} \sum_{c=1}^K \sum_{i \in T_c} \mathbf{1}\{\widehat{y}_h(x_i) \neq c\}, \qquad u = \sum_{c=1}^K u_c.$$

### K.1.1    Pairwise margin losses

For each ordered pair $(c, r)$ with $r \neq c$, define the scalar pairwise margin

$$f_{c,r,h}(x) = h_c(x) - h_r(x).$$

For $\gamma > 0$, define the ramp loss

$$\psi_\gamma(t) = \min\{1, \max\{0, 1 - t/\gamma\}\},$$

and

$$\ell_{c,r}^\gamma(h, x) = \psi_\gamma(f_{c,r,h}(x)).$$

The loss $\psi_\gamma$ is $1/\gamma$-Lipschitz and takes values in $[0, 1]$. For a node with true class $c$,

$$\mathbf{1}\{\widehat{y}_h(x) \neq c\} \leq \sum_{r \neq c} \mathbf{1}\{h_r(x) \geq h_c(x)\}$$

$$= \sum_{r \neq c} \mathbf{1}\{f_{c,r,h}(x) \leq 0\}$$

$$\leq \sum_{r \neq c} \psi_\gamma(f_{c,r,h}(x)) = \sum_{r \neq c} \ell_{c,r}^\gamma(h, x).$$

Thus the multi-class error is upper bounded by a sum of scalar pairwise margin losses.

Define

$$\widehat{L}_{m_c}^{c,r,\gamma}(h) = \frac{1}{m_c} \sum_{i \in S_c} \ell_{c,r}^\gamma(h, x_i), \qquad L_{u_c}^{c,r,\gamma}(h) = \frac{1}{u_c} \sum_{i \in T_c} \ell_{c,r}^\gamma(h, x_i). \qquad (14)$$

Also define the scalar pairwise score class

$$\mathcal{F}_{c,r} = \{x \mapsto h_c(x) - h_r(x) : h \in \mathcal{H}\}.$$

**Theorem K.1** (Multi-class transductive bound)**.** *Let*

$$Q_c = \frac{1}{m_c} + \frac{1}{u_c}, \qquad S_c = \frac{m_c + u_c}{(m_c + u_c - 1/2)(1 - 1/(2\max\{m_c, u_c\}))}, \tag{15}$$

*and let $c_0 = \sqrt{32\ln(4e)/3}$. Let $C_\psi$ denote the scalar contraction constant for the ramp loss; one can take $C_\psi = 1$ under a centered contraction convention and $C_\psi = 2$ under the most common off-the-shelf contraction statement. Then with probability at least $1 - \delta$ over the class-wise transductive split, the following holds simultaneously for all $h \in \mathcal{H}$:*

$$L_u^K(h) \le \sum_{c=1}^{K} \frac{u_c}{u} \sum_{r \ne c} \left[ \widehat{L}_{m_c}^{c,r,\gamma}(h) + \frac{C_\psi}{\gamma} \mathfrak{R}_{m_c,u_c}(\mathcal{F}_{c,r}) + c_0 Q_c \sqrt{\min\{m_c, u_c\}} \right. \tag{16}$$
$$\left. + \sqrt{\frac{S_c Q_c}{2} \log \frac{K(K-1)}{\delta}} \right].$$

*Proof.* Fix $(c,r)$ with $r \ne c$. Apply the scalar transductive Rademacher bound of El-Yaniv & Pechyony (2009) to the bounded loss class

$$\mathcal{L}_{c,r}^\gamma = \{i \mapsto \ell_{c,r}^\gamma(h, x_i) : h \in \mathcal{H}, \ i \in I_c\}.$$

With probability at least $1 - \delta_{c,r}$, for all $h \in \mathcal{H}$,

$$L_{u_c}^{c,r,\gamma}(h) \le \widehat{L}_{m_c}^{c,r,\gamma}(h) + \mathfrak{R}_{m_c,u_c}(\mathcal{L}_{c,r}^\gamma) + c_0 Q_c \sqrt{\min\{m_c, u_c\}} + \sqrt{\frac{S_c Q_c}{2} \log \frac{1}{\delta_{c,r}}}.$$

By scalar contraction for the $1/\gamma$-Lipschitz ramp loss,

$$\mathfrak{R}_{m_c,u_c}(\mathcal{L}_{c,r}^\gamma) \le \frac{C_\psi}{\gamma} \mathfrak{R}_{m_c,u_c}(\mathcal{F}_{c,r}).$$

Set $\delta_{c,r} = \delta/(K(K-1))$ and take a union bound over the $K(K-1)$ ordered class pairs. Finally, use

$$\mathbf{1}\{\widehat{y}_h(x_i) \ne c\} \le \sum_{r \ne c} \ell_{c,r}^\gamma(h, x_i)$$

for each $i \in T_c$ and average over classes with weights $u_c/u$. $\qquad\square$

### K.1.2   GNN complexity term in the multi-class bound

By subadditivity of the supremum,

$$\mathfrak{R}_{m_c,u_c}(\mathcal{F}_{c,r}) = \mathfrak{R}_{m_c,u_c}(\{h_c - h_r : h \in \mathcal{H}\})$$
$$\le \mathfrak{R}_{m_c,u_c}(\mathcal{H}_c) + \mathfrak{R}_{m_c,u_c}(\mathcal{H}_r).$$

If each coordinate class has the same GNN complexity bound, then

$$\mathfrak{R}_{m_c,u_c}(\mathcal{F}_{c,r}) \le 2 C_d B_f \|G_f(A)[c]\|_\infty^{2(d-1)} \prod_{j=1}^{d} U_F(j) \sqrt{\frac{m_c + u_c}{m_c u_c}}.$$

For the symmetric normalized GCN filter,

$$\|G_f(A)[c]\|_\infty^{2(d-1)} \le \left( \frac{\deg_c^{\max} + 1}{\deg^{\min} + 1} \right)^{d-1}.$$

Therefore, suppressing constants not involving sample size, the multi-class bound contains the class-wise term

$$(K-1)\left(\frac{\deg_c^{\max}+1}{\deg^{\min}+1}\right)^{d-1}\sqrt{\frac{m_c+u_c}{m_c u_c}}.$$

The conservative $(K-1)$ factor is the price of the pairwise reduction. This is acceptable for a response because the purpose is to remove the theory/experiment mismatch without claiming a sharper theorem than we can prove.

### K.2  Understanding pseudo-labeling Effect

The original clean-label theorem counts genuinely labeled nodes. UPL adds adaptively selected pseudo-labels, so pseudo-labels should not be inserted into $m_c$ as if they were clean labels. The clean way to connect UPL to the theory is to separate three quantities:

1. the number of selected pseudo-labels assigned to a minority class;

2. the fraction of those selected pseudo-labels that are actually correct; and

3. the mismatch between the selected pseudo-labeled nodes and the remaining class-conditional evaluation nodes.

In the following, we provide result on PL performance in UPL provided that the UPL selects sufficiently accurate and sufficiently representative pseudo-labels, then its effect appears through an effective sample-size term, a pseudo-label noise term, and a selection/coverage term.

Let $\mathcal{L}_c$ be the clean labeled training nodes of true class $c$, with $|\mathcal{L}_c| = m_c$. At pseudo-labeling iteration $t$, the model produces probabilities $\widehat{p}_i \in \Delta^{K-1}$ and pseudo-labels

$$\widetilde{y}_i = \arg\max_{a\in[K]}\widehat{p}_{i,a}, \qquad \widehat{q}_i = \max_{a\in[K]}\widehat{p}_{i,a}.$$

Let $U_i$ be the uncertainty/stability score computed from graph perturbations. In the current manuscript this score is entropy variance, but for the theory it is better to define $U_i$ abstractly; it can later be instantiated as entropy variance, prediction-vector variance, class-agreement rate, variation ratio, or an average pairwise divergence score.

If $\mathcal{C}_{\min}$ is the set of minority classes, the UPL selection mask can be written as

$$M_i^{(t)} = \mathbf{1}\{i\notin S_{\text{clean}}\}\mathbf{1}\{\widetilde{y}_i\in\mathcal{C}_{\min}\}\mathbf{1}\{\eta_l\le\widehat{q}_i\le\eta_u\}\mathbf{1}\{U_i\le q_\alpha\}.$$

For each predicted class $c$, define

$$\widehat{S}_{c,t}^{\text{PL}} = \{i\in\mathcal{U}: M_i^{(t)}=1,\ \widetilde{y}_i=c\}, \qquad a_{c,t} = |\widehat{S}_{c,t}^{\text{PL}}|.$$

Here $a_{c,t}$ is the number of nodes selected by UPL and *assigned* pseudo-label $c$. It is not necessarily the number of truly class-$c$ nodes selected.

For analysis only, define the selected pseudo-label error rate

$$\epsilon_{c,t} = \frac{1}{a_{c,t}}\sum_{i\in\widehat{S}_{c,t}^{\text{PL}}}\mathbf{1}\{y_i\ne c\}, \qquad \epsilon_{c,t}=0 \quad\text{if } a_{c,t}=0.$$

Since every node in $\widehat{S}_{c,t}^{\text{PL}}$ is assigned pseudo-label $c$, the event $y_i\ne c$ is exactly the event that the pseudo-label is wrong. Define the correctly selected subset and its size by

$$\widehat{S}_{c,t}^{\text{cor}} = \{i\in\widehat{S}_{c,t}^{\text{PL}}: y_i=c\}, \qquad a_{c,t}^{\text{cor}} = |\widehat{S}_{c,t}^{\text{cor}}| = a_{c,t}(1-\epsilon_{c,t}).$$

The effective number of clean class-$c$ examples after pseudo-labeling is therefore

$$n_{c,t}^{\text{eff}} = m_c + a_{c,t}^{\text{cor}} = m_c + a_{c,t}(1 - \epsilon_{c,t}).$$

This is the quantity that should enter the clean-label sample-size term. The larger apparent count $m_c + a_{c,t}$ is justified only when $\epsilon_{c,t}$ is small.

Let

$$\widehat{S}_t^{\text{PL}} = \bigcup_{c=1}^{K} \widehat{S}_{c,t}^{\text{PL}}, \qquad \mathcal{R}_t = \mathcal{U} \setminus \widehat{S}_t^{\text{PL}},$$

and define the remaining true class-$c$ evaluation nodes by

$$\mathcal{R}_{c,t} = \{i \in \mathcal{R}_t : y_i = c\}, \qquad u_{c,t}^{\text{rem}} = |\mathcal{R}_{c,t}|, \qquad u_t^{\text{rem}} = \sum_{c=1}^{K} u_{c,t}^{\text{rem}}.$$

The true labels in these definitions are not available to the algorithm; they are analysis quantities.

### K.2.1 Pairwise losses and class-wise risk decomposition

For an ordered class pair $(c, r)$ with $r \neq c$, define

$$f_{c,r,h}(x) = h_c(x) - h_r(x)$$

and the ramp margin loss

$$\ell_\gamma(s) = \begin{cases} 1, & s \leq 0, \\ 1 - s/\gamma, & 0 < s < \gamma, \\ 0, & s \geq \gamma. \end{cases}$$

We abbreviate $\ell_{c,r}^\gamma(h, i) = \ell_\gamma(f_{c,r,h}(x_i))$ and write

$$\mathcal{F}_{c,r} = \{x \mapsto h_c(x) - h_r(x) : h \in \mathcal{H}\}.$$

The remaining multi-class error is

$$L_{\mathcal{R}_t}^{\text{mc}}(h) = \frac{1}{u_t^{\text{rem}}} \sum_{i \in \mathcal{R}_t} \mathbf{1}\{\widehat{y}_h(x_i) \neq y_i\}, \qquad \widehat{y}_h(x_i) = \arg\max_{a \in [K]} h_a(x_i).$$

It decomposes by true class as

$$L_{\mathcal{R}_t}^{\text{mc}}(h) = \sum_{c=1}^{K} \frac{u_{c,t}^{\text{rem}}}{u_t^{\text{rem}}} L_{\mathcal{R}_{c,t}}^{\text{mc}}(h),$$

where

$$L_{\mathcal{R}_{c,t}}^{\text{mc}}(h) = \frac{1}{u_{c,t}^{\text{rem}}} \sum_{i \in \mathcal{R}_{c,t}} \mathbf{1}\{\widehat{y}_h(x_i) \neq c\}.$$

For each pair $(c, r)$, define

$$L_{\mathcal{R}_{c,t}}^{c,r,\gamma}(h) = \frac{1}{u_{c,t}^{\text{rem}}} \sum_{i \in \mathcal{R}_{c,t}} \ell_{c,r}^\gamma(h, i).$$

If $\widehat{y}_h(x_i) \neq c$, then at least one rival class $r \neq c$ has $h_r(x_i) \geq h_c(x_i)$; hence the corresponding ramp loss is one. Therefore

$$L_{\mathcal{R}_{c,t}}^{\text{mc}}(h) \leq \sum_{r \neq c} L_{\mathcal{R}_{c,t}}^{c,r,\gamma}(h).$$

This is the pairwise reduction used in the proof.

### K.2.2 Correctly selected versus pseudo-labeled empirical losses

There are two empirical losses to distinguish. The first is the ideal empirical loss on the clean labeled nodes plus the *correctly selected* pseudo-labeled nodes:

$$\widehat{L}_{\text{cor}}^{c,r,\gamma}(h) = \frac{1}{m_c + a_{c,t}^{\text{cor}}} \left( \sum_{i \in \mathcal{L}_c} \ell_{c,r}^{\gamma}(h,i) + \sum_{i \in \widehat{S}_{c,t}^{\text{cor}}} \ell_{c,r}^{\gamma}(h,i) \right).$$

This quantity is an oracle analysis object because it uses knowledge of which accepted pseudo-labels are correct.

The second is the empirical loss actually induced by treating all nodes in $\widehat{S}_{c,t}^{\text{PL}}$ as pseudo-labeled class-$c$ nodes:

$$\widehat{L}_{\text{PL}}^{c,r,\gamma}(h; \widetilde{y}) = \frac{1}{m_c + a_{c,t}} \left( \sum_{i \in \mathcal{L}_c} \ell_{c,r}^{\gamma}(h,i) + \sum_{i \in \widehat{S}_{c,t}^{\text{PL}}} \ell_{c,r}^{\gamma}(h,i) \right).$$

The $\widetilde{y}$ indicates that the selected set is defined by pseudo-labels, although the displayed pairwise loss is written with target class $c$.

**Lemma K.2** (Pseudo-label noise penalty for pairwise ramp losses)**.** *For any $c$, $r \neq c$, and $h$, the correctly selected empirical loss is bounded by the pseudo-labeled empirical loss plus the selected pseudo-label error fraction:*

$$\widehat{L}_{\text{cor}}^{c,r,\gamma}(h) \leq \widehat{L}_{\text{PL}}^{c,r,\gamma}(h; \widetilde{y}) + \frac{a_{c,t}}{m_c + a_{c,t}} \epsilon_{c,t}.$$

*Proof.* Write $a = a_{c,t}$, $b = a_{c,t}^{\text{cor}}$, and $m = m_c$. Since $b = a(1 - \epsilon_{c,t})$, the number of incorrectly selected pseudo-labels assigned to class $c$ is $a - b = a\epsilon_{c,t}$. Let

$$S_0 = \sum_{i \in \mathcal{L}_c} \ell_{c,r}^{\gamma}(h,i), \qquad S_1 = \sum_{i \in \widehat{S}_{c,t}^{\text{cor}}} \ell_{c,r}^{\gamma}(h,i),$$

and

$$S_2 = \sum_{i \in \widehat{S}_{c,t}^{\text{PL}} \setminus \widehat{S}_{c,t}^{\text{cor}}} \ell_{c,r}^{\gamma}(h,i).$$

Because the ramp loss lies in $[0, 1]$, we have $S_0 + S_1 \leq m + b$ and $S_2 \geq 0$. By definition,

$$\widehat{L}_{\text{cor}}^{c,r,\gamma}(h) = \frac{S_0 + S_1}{m + b}, \qquad \widehat{L}_{\text{PL}}^{c,r,\gamma}(h; \widetilde{y}) = \frac{S_0 + S_1 + S_2}{m + a}.$$

Therefore

$$
\begin{aligned}
\widehat{L}_{\text{cor}}^{c,r,\gamma}(h) - \widehat{L}_{\text{PL}}^{c,r,\gamma}(h; \widetilde{y}) &= \frac{(S_0 + S_1)(m + a) - (S_0 + S_1 + S_2)(m + b)}{(m + b)(m + a)} \\
&= \frac{(S_0 + S_1)(a - b) - S_2(m + b)}{(m + b)(m + a)} \\
&\leq \frac{(m + b)(a - b)}{(m + b)(m + a)} = \frac{a - b}{m + a} = \frac{a}{m + a} \epsilon_{c,t}.
\end{aligned}
$$

Rearranging gives the claim. The case $a = 0$ is included by the convention $\epsilon_{c,t} = 0$. □

### K.2.3 Coverage assumption for adaptive pseudo-label selection

The scalar transductive Rademacher bound, Theorem 4.6, applies directly to a clean labeled/unlabeled transductive split. UPL, however, selects pseudo-labels adaptively using the trained model, confidence thresholds, and uncertainty filtering. Therefore, we include a coverage term rather than pretending that the selected pseudo-labels are a fresh random class-wise sample.

**Assumption K.3** (Conditional class-wise coverage)**.** Condition on the UPL-selected sets at iteration $t$. For each class $c$, ordered pair $(c, r)$, and hypothesis $h$, suppose the remaining class-$c$ margin loss satisfies

$$L_{\mathcal{R}_{c,t}}^{c,r,\gamma}(h) \leq \widehat{L}_{\mathrm{cor}}^{c,r,\gamma}(h) + \frac{C_\psi}{\gamma}\mathfrak{R}_{n_{c,t}^{\mathrm{eff}}, u_{c,t}^{\mathrm{rem}}}(\mathcal{F}_{c,r}) + \Gamma_{c,t}(\delta) + \Delta_{c,t}^r(h).$$

Here $\Delta_{c,t}^r(h) \geq 0$ measures the extra mismatch caused by adaptive selection. If the correctly selected pseudo-labeled nodes are representative of the remaining true class-$c$ nodes under the pairwise margin loss, this term is small. If UPL selects only very easy, redundant, or topologically atypical nodes, this term can be large.

Assumption K.3 is the place where selection bias enters. In the idealized case where the correctly selected pseudo-labeled class-$c$ nodes and the remaining class-$c$ nodes behave like a class-wise transductive split, $\Delta_{c,t}^r(h)$ can be viewed as negligible and the displayed inequality reduces to the scalar transductive Rademacher comparison. A concrete way to think about $\Delta_{c,t}^r(h)$ is as an integral-probability-metric-style discrepancy between correctly selected nodes and remaining nodes over the margin-loss class. For example, one may upper-bound it by

$$\sup_{g \in \mathcal{G}_{c,r}} \left| \frac{1}{u_{c,t}^{\mathrm{rem}}} \sum_{i \in \mathcal{R}_{c,t}} g(x_i) - \frac{1}{a_{c,t}^{\mathrm{cor}}} \sum_{i \in \widehat{S}_{c,t}^{\mathrm{cor}}} g(x_i) \right|,$$

where $\mathcal{G}_{c,r} = \{x \mapsto \ell_\gamma(f_{c,r,h}(x), y) : h \in \mathcal{H}\}$ and the second average is defined conservatively when $a_{c,t}^{\mathrm{cor}} = 0$.

**Proposition K.4** (UPL-style class-wise bound, conditional on pseudo-label quality and coverage)**.** *Condition on the UPL-selected pseudo-labeled sets at iteration $t$ and suppose Assumption K.3 holds. Let*

$$n_{c,t}^{\mathrm{eff}} = m_c + a_{c,t}^{\mathrm{cor}} = m_c + a_{c,t}(1 - \epsilon_{c,t}).$$

*Define*

$$Q_{c,t} = \frac{1}{n_{c,t}^{\mathrm{eff}}} + \frac{1}{u_{c,t}^{\mathrm{rem}}}$$

*and*

$$S_{c,t} = \frac{n_{c,t}^{\mathrm{eff}} + u_{c,t}^{\mathrm{rem}}}{(n_{c,t}^{\mathrm{eff}} + u_{c,t}^{\mathrm{rem}} - 1/2)\left(1 - 1/(2\max\{n_{c,t}^{\mathrm{eff}}, u_{c,t}^{\mathrm{rem}}\})\right)}.$$

*Let*

$$\Gamma_{c,t}(\delta) = c_0 Q_{c,t}\sqrt{\min\{n_{c,t}^{\mathrm{eff}}, u_{c,t}^{\mathrm{rem}}\}} + \sqrt{\frac{S_{c,t}Q_{c,t}}{2}\log\frac{K(K-1)}{\delta}}.$$

*Then the remaining-node multi-class error satisfies*

$$L_{\mathcal{R}_t}^{\mathrm{mc}}(h) \leq \sum_{c=1}^{K} \frac{u_{c,t}^{\mathrm{rem}}}{u_t^{\mathrm{rem}}} B_{c,t}(h) \tag{17}$$

*with the class-$c$ bound term*

$$
\begin{aligned}
B_{c,t}(h) = \sum_{r \neq c} \Bigg[ &\widehat{L}_{\mathrm{PL}}^{c,r,\gamma}(h; \widetilde{y}) + \frac{a_{c,t}}{m_c + a_{c,t}}\epsilon_{c,t} + \Delta_{c,t}^r(h) \\
&+ \frac{C}{\gamma}\|G_f(A)[c]\|_\infty^{2(d-1)} \prod_{j=1}^{d} U_F(j)\sqrt{\frac{n_{c,t}^{\mathrm{eff}} + u_{c,t}^{\mathrm{rem}}}{n_{c,t}^{\mathrm{eff}} u_{c,t}^{\mathrm{rem}}}} + \Gamma_{c,t}(\delta) \Bigg],
\end{aligned}
\tag{18}
$$

*where $C$ hides architecture-dependent constants independent of $(m_c, a_{c,t}, u_{c,t}^{\mathrm{rem}})$.*

*Proof.* We prove the claim in four steps.

*Step 1: decompose the remaining multi-class error by true class.* By definition,

$$L_{\mathcal{R}_t}^{\mathrm{mc}}(h) = \sum_{c=1}^{K} \frac{u_{c,t}^{\mathrm{rem}}}{u_t^{\mathrm{rem}}} L_{\mathcal{R}_{c,t}}^{\mathrm{mc}}(h).$$

*Step 2: upper bound the class-c error by pairwise margin losses.* For every node $i$ with true class $c$,

$$\mathbf{1}\{\widehat{y}_h(x_i) \neq c\} \leq \sum_{r \neq c} \mathbf{1}\{h_r(x_i) \geq h_c(x_i)\}$$

$$\leq \sum_{r \neq c} \ell_\gamma(h_c(x_i) - h_r(x_i)) = \sum_{r \neq c} \ell_{c,r}^\gamma(h, i).$$

Averaging over $i \in \mathcal{R}_{c,t}$ gives

$$L_{\mathcal{R}_{c,t}}^{\mathrm{mc}}(h) \leq \sum_{r \neq c} L_{\mathcal{R}_{c,t}}^{c,r,\gamma}(h).$$

*Step 3: apply the conditional scalar transductive comparison.* By Assumption K.3, for every ordered pair $(c, r)$,

$$L_{\mathcal{R}_{c,t}}^{c,r,\gamma}(h) \leq \widehat{L}_{\mathrm{cor}}^{c,r,\gamma}(h) + \frac{C_\psi}{\gamma} \mathfrak{R}_{n_{c,t}^{\mathrm{eff}}, u_{c,t}^{\mathrm{rem}}}(\mathcal{F}_{c,r}) + \Gamma_{c,t}(\delta) + \Delta_{c,t}^r(h).$$

The union bound over the $K(K-1)$ ordered pairs is the reason for the factor $\log(K(K-1)/\delta)$ in $\Gamma_{c,t}(\delta)$.

The pairwise GNN complexity calculation gives

$$\mathfrak{R}_{n_{c,t}^{\mathrm{eff}}, u_{c,t}^{\mathrm{rem}}}(\mathcal{F}_{c,r}) \leq C' \|G_f(A)[c]\|_\infty^{2(d-1)} \prod_{j=1}^{d} U_F(j) \sqrt{\frac{n_{c,t}^{\mathrm{eff}} + u_{c,t}^{\mathrm{rem}}}{n_{c,t}^{\mathrm{eff}} u_{c,t}^{\mathrm{rem}}}}.$$

Absorbing $C_\psi C'$ into $C$ yields the complexity term in the proposition.

*Step 4: replace the ideal correctly selected empirical loss by the actual pseudo-labeled empirical loss.* Lemma K.2 gives

$$\widehat{L}_{\mathrm{cor}}^{c,r,\gamma}(h) \leq \widehat{L}_{\mathrm{PL}}^{c,r,\gamma}(h; \widetilde{y}) + \frac{a_{c,t}}{m_c + a_{c,t}} \epsilon_{c,t}.$$

Substituting this inequality into Step 3, summing over $r \neq c$, and then substituting the result into the class decomposition from Steps 1–2 gives

$$L_{\mathcal{R}_t}^{\mathrm{mc}}(h) \leq \sum_{c=1}^{K} \frac{u_{c,t}^{\mathrm{rem}}}{u_t^{\mathrm{rem}}} B_{c,t}(h),$$

which proves the proposition. □

*Remark* K.5 (Interpretation of the three UPL-dependent terms). Proposition K.4 separates the UPL effect into three quantities:

$$\underbrace{\sqrt{\frac{1}{m_c + a_{c,t}(1 - \epsilon_{c,t})} + \frac{1}{u_{c,t}^{\mathrm{rem}}}}}_{\text{effective sample-size term}} \quad + \quad \underbrace{\frac{a_{c,t}}{m_c + a_{c,t}} \epsilon_{c,t}}_{\text{pseudo-label noise}} \quad + \quad \underbrace{\Delta_{c,t}^r(h)}_{\text{selection/coverage mismatch}} \quad .$$

The first term explains the intended benefit of minority pseudo-labeling: reliable accepted labels increase the effective class-$c$ training count. The second term explains why pseudo-labels cannot be treated as clean labels. If the uncertainty filter accepts many incorrect pseudo-labels, $\epsilon_{c,t}$ is large, the effective sample size barely increases, and the noise penalty can dominate. The third term explains why selecting only easy, redundant, or topologically atypical nodes may not help the remaining evaluation nodes, even when many selected pseudo-labels are correct.

*Remark* K.6 (How uncertainty-aware selection affects). The uncertainty-aware part of UPL should be linked to $\epsilon_{c,t}$ and $\Delta_{c,t}^r(h)$. The lower confidence threshold, upper confidence threshold, minority-only rule, and uncertainty threshold jointly determine $a_{c,t}$, $\epsilon_{c,t}$, and $\Delta_{c,t}^r(h)$.

### K.3   Understanding uncertainty-aware selection Effect

Adding selected pseudo-labels can reduce the sample-size term, but only if the pseudo-label error term remains small. SER and the uncertainty threshold are designed to keep $\epsilon_{c,t}$ small by accepting predictions stable under graph perturbations. The band-pass rule and minority-only rule control the number and distribution of selected pseudo-labels, thereby affecting both $a_{c,t}$ and $\Delta_{c,t}^r(h)$.

### K.3.1   Selection probability, noise, and the trade-off

For thresholds $(\eta_l, \eta_u, q_\alpha)$, define the conditional pseudo-label error rate

$$\epsilon_c(\eta_l, \eta_u, q_\alpha) := \mathbb{P}\Big(\widetilde{Y} \neq Y \,\big|\, \widetilde{Y} = c, c \in \mathcal{C}_{\min}, \eta_l \leq \widehat{q} \leq \eta_u, U \leq q_\alpha\Big),$$

and the expected selected count

$$a_c(\eta_l, \eta_u, q_\alpha) := u_c \, \mathbb{P}\Big(\widetilde{Y} = c, \eta_l \leq \widehat{q} \leq \eta_u, U \leq q_\alpha \,\big|\, Y = c\Big).$$

Then the UPL selection rule can be explained as trying to improve the class-$c$ term

$$\Phi_c \approx A_c \sqrt{\frac{1}{m_c + a_c(1 - \epsilon_c)} + \frac{1}{u_c^{\text{rem}}}} + \frac{a_c}{m_c + a_c}\epsilon_c + \Delta_c,$$

where $A_c$ contains the graph-filter and architecture constants. The first term favors selecting more correct minority pseudo-labels; the second penalizes noisy pseudo-labels; the third penalizes distributional mismatch/selection bias.

