# OpenReview forum: "UPL: Uncertainty-aware Pseudo-labeling for Imbalanced Transductive Node Classification"
_TMLR — Under review for TMLR_

### Review · Reviewer_dQyD · 2026-06-14

**Summary Of Contributions:**

## Summary
This paper introduces class imbalance in transductive node classification on graphs. The main contributions are threefold: (1) a theoretical upper bound on population risk for imbalanced transductive node classification using Rademacher complexity analysis. This shows that the number of minority class samples mainly influences the bound. (2) an algorithm called UPL, which includes a Selective Edge Removal (SER) uncertainty estimator, a dual-threshold band-pass pseudo-label filter, and a Balanced Softmax loss, all focused on minority classes. (3) an empirical evaluation that covers six benchmark datasets, both homophilic and heterophilic, along with four imbalance ratios, including ablation studies and comparisons with multiple baselines.

## Key Strengths
- The theoretical motivation matches the algorithm design. Theorem 4.6 clearly shows that the population risk upper bound is influenced by the minority class sample size. This finding directly supports the minority-only pseudo-labeling strategy in UPL. This connection between theory and algorithm is a real strength.

- The Selective Edge Removal (SER) mechanism is a clear and efficient way to estimate uncertainty while considering topology. Measuring entropy variance across degree-biased graph changes makes sense due to GNN message-passing behavior.

- The experimental scope is broad, including four GNN backbones, four imbalance ratios, and nine datasets, such as large-scale Flickr, with averages over ten repetitions and standard errors reported.

## Key Weaknesses
- The theoretical analysis focuses on binary classification with a $\gamma$-margin loss. However, since all experiments involve multi-class settings with up to 10 classes, there is a significant gap between theory and practice. The bound in Theorem 4.6 cannot be directly used to explain the experimental results as stated.

- The ablation study shows a troubling trend: the Uncertainty component by itself, without Balanced Softmax or pseudo-labeling, produces results that are the same as Vanilla across several datasets. For example, on Cora, the balanced accuracy is 53.28 for both. This indicates that the SER uncertainty block does not contribute anything on its own and only shows improvements when combined with the other components. Therefore, the individual impact of the uncertainty mechanism is unclear.

- The upper threshold $\eta_u$, which is a key innovation known as the "band-pass" design, is not compared to simpler options like confidence-only thresholding with $\eta_l$ alone. The ablation study uses BS + PL, which implicitly includes the band-pass filter, but does not specifically test the effect of $\eta_u$. Thus, it is unclear whether the band-pass design is necessary or if $\eta_l$ alone is enough.

**Audience:**

Yes

**Audience Explanation:**

Imbalanced node classification is a real problem, and pseudo-labeling in the transductive GNN setting has not been fully explored. The link between the theoretical risk bound and the algorithm's design is important to discuss, even if the theory requires more development.

**Claims And Evidence:**

No

**Claims Explanation:**

The claim of top performance stands for most homophilic datasets and many heterophilic ones, but there are important exceptions, such as Squirrel GCN and Cora/Citeseer GAT compared to UNREAL. The assertion that SER provides topology-aware uncertainty to improve pseudo-label quality is not demonstrated by the ablation alone, and the block shows no benefit when used on its own. The theoretical assertion about convergence rate and the dominance of the error bound in the minority class is valid within its specified binary classification context, but cannot formally support the multi-class experiments.

**Requested Changes:**

- I would like to ask authors to extend the theory to multi-class settings or clearly define the claims. All experiments involve multiple classes. Either extend Theorem 4.6 to the multi-class case using a one-vs-rest or margin-based multi-class approach, or explicitly limit empirical claims to contexts where the binary theory applies. The current gap between theory and experiment undermines the paper's main argument.

- The band-pass design is highlighted as an important innovation, but the ablation does not include a scenario with only $\eta_l$ (without $\eta_u$). Please add an ablation condition using a standard single lower-threshold pseudo-labeling approach and compare it directly to the dual-threshold version. This is needed to support the band-pass framing.

- I suggest that authors clarify the independent role of SER. Table 3 shows that "+ Uncertainty" by itself matches Vanilla exactly on several datasets. The paper should either (a) explain why SER does not have an effect without pseudo-labeling, since it only filters the pseudo-label set, which is empty without PL, making this clear in the text, or (b) evaluate SER independently in a context where pseudo-labels are created without the uncertainty filter and compare it to UPL. As it stands, a reader may wrongly think that SER adds no value.

- I would like to ask authors to compare against UNREAL and BAT more thoroughly on heterophilic datasets. On GCN Squirrel, UNREAL outperforms UPL in both bAcc and F1. The paper states that UPL "consistently delivers better results in most cases" for heterophilic data, but Table 2 does not fully support this. Adjust the claim to accurately reflect where UPL does and does not outperform. Provide a more detailed discussion of the Squirrel result, specifically.

- The main Tables 1 and 2 include standard errors, but for close comparisons (e.g., UPL 77.50 ± 0.44 vs. UNREAL 75.35 ± 1.41 on PubMed, and UPL 43.34 ± 0.25 vs. UNREAL 45.65 ± 1.08 on Squirrel), it is unclear from standard error alone whether the differences are statistically significant. I suggest that authors report p-values or confidence intervals for key pairwise comparisons, especially where UPL claims an advantage over the nearest competitor.

- The UPL algorithm repeats pseudo-label assignment and retraining for k rounds. There is no theoretical or empirical analysis of when this process converges or stabilizes. Figure 2 shows sensitivity to threshold values but not to the number of iterations. A plot of performance versus pseudo-labeling iteration count for representative datasets would clarify the practical stopping point and whether k = 10 is always enough.

---

### Review · Reviewer_RXZi · 2026-06-23

**Summary Of Contributions:**

This paper studies class-imbalanced transductive node classification and proposes UPL, a pseudo-labeling framework combining minority-class augmentation, dual confidence thresholds, graph-perturbation-based uncertainty filtering, and a class-balancing loss. The method is evaluated on several homophilic and heterophilic graph datasets and generally achieves competitive results.

The problem is relevant, and the proposed framework is simple and potentially useful. However, the main theoretical result is not established by the current proof, the proposed uncertainty score does not reliably measure prediction consistency, and the experiments do not directly validate the claimed improvement in pseudo-label quality. Several important method definitions are also inaccurate or insufficiently specified.

**Audience:**

Yes

**Audience Explanation:**

Class imbalance in graph learning is an important problem, and a lightweight pseudo-labeling method that can be combined with standard GNN backbones may interest researchers in graph learning and long-tailed recognition. The observation that minority-only pseudo-labeling can be effective is also potentially useful.

**Claims And Evidence:**

No

**Claims Explanation:**

The experiments support the narrower claim that UPL is competitive on several imbalanced node-classification benchmarks. However, the paper’s stronger theoretical and methodological claims are not adequately supported.

Most importantly, Proposition 4.5 contains algebraic errors that affect the dependence of the Rademacher-complexity bound on the sample sizes. Consequently, Theorem 4.6 and the stated convergence rate are not well-established by the current proof.

In addition, the proposed uncertainty score is the variance of predictive entropy across perturbed graphs. Equal entropy does not imply equal predictions: two perturbations may produce opposite predicted classes while having identical entropy and therefore zero entropy variance. Thus, the score does not generally measure prediction consistency.

Finally, the current ablations report final classification performance but do not directly show that the uncertainty filter improves pseudo-label accuracy, calibration, or error detection. Therefore, the empirical evidence does not fully support the central claims about uncertainty-aware pseudo-labeling.

**Requested Changes:**

Please address the following issues:

1. Correct the main theoretical result.

In Proposition 4.5, since
$
\mathbb{E}[\epsilon_i^2]=2p,
$
the relevant second-moment term scales with $\sqrt{2p}$, not $2p$. The proof also introduces an unsupported $p^2$factor in the bounded-difference calculation. These errors affect the rate, rather than only a constant. Theorem 4.6 and the claimed $O(1/u)$ convergence rate should therefore be rederived. The probability statement in Proposition 4.4 should also account for the joint class-wise events, for example through a union bound.

2. Revise the uncertainty definition and associated claims.

Entropy variance does not detect changes in the predicted class. For example,

$
(0.99,0.01) \quad\text{and}\quad (0.01,0.99)
$

have identical entropy but opposite predictions. The authors should use or compare against measures that retain class information, such as prediction-vector variance, class-agreement rate, variation ratio, or pairwise divergence. Claims that the method captures both aleatoric and epistemic uncertainty should be removed unless they are formally and empirically justified.

3. Clarify the connection between the theory and UPL.

The theoretical quantity $m_i$ counts genuinely labeled samples, whereas UPL adds adaptively selected and potentially incorrect pseudo-labels. These pseudo-labels cannot be treated as additional clean labels without accounting for label noise and selection bias.

4. Directly evaluate the uncertainty and pseudo-labeling components.

The current ablations do not establish that SER consistently improves pseudo-label quality. The revised paper should report pseudo-label precision, the number and class distribution of selected nodes, error rates among accepted and rejected nodes, and uncertainty-based error-detection metrics such as AUC. Calibration metrics or risk-coverage curves would also strengthen the claims. Comparisons with random edge removal and alternative uncertainty scores are needed to isolate the contribution of SER.

5. Correct the method description and improve reproducibility.

Equation (5) is a class-weighted cross-entropy loss rather than the standard Balanced Softmax objective cited by the paper. Which loss is implemented in your experiment setup?

---

### Review · Reviewer_vVsF · 2026-06-24

**Summary Of Contributions:**

The paper addresses class-imbalanced transductive node classification through two main contributions: a transductive-Rademacher population-risk bound for GCN-style models, linking generalization to class-specific sample sizes and graph-degree properties, and UPL, an uncertainty-aware pseudo-labeling method combining minority-focused selection, band-pass confidence thresholds, SER uncertainty estimation via entropy variance under graph perturbations, and balanced-softmax-style training.

Experiments cover homophilic and heterophilic benchmarks, multiple imbalance ratios, GCN/GAT/GraphSAGE backbones, bAcc/F1 over repeated runs, and runtime comparison with UNREAL. Strengths include the theory-to-method motivation, the simplicity of SER, and broad evaluation. Weaknesses include the binary theory versus multi-class experiments, insufficient isolation of the SER contribution in ablations, and some overclaimed comparisons where UPL is not uniformly best.

**Additional Comments:**

No

**Audience:**

Yes

**Audience Explanation:**

Yes. Imbalanced node classification and self-training/calibration in GNNs are active subareas;

**Claims And Evidence:**

No

**Claims Explanation:**

Main reasons:



1. **Theory-to-experiment mismatch**

   The theory is binary$(\gamma)$-margin based, but experiments are multi-class. The authors later acknowledge that extending to multi-class is future work, so the theory does not fully support the multi-class empirical claims.



2. **Blanket superiority is not fully supported**

   The paper claims superior performance, but the tables include cases where UPL is not best. For example, on Amazon Photo under GCN, GraphENS+TAM outperforms UPL.



3. **Uncertainty contribution is not isolated enough**

   The paper proposes SER as a topology-aware uncertainty estimator, but the ablation does not cleanly hold pseudo-labeling fixed while varying only the uncertainty estimator. So the evidence for SER specifically is not fully convincing.

**Requested Changes:**

1. The band-pass confidence rule is reasonable but heuristic. The paper includes a limited threshold sweep in Figure 2, but this does not fully establish robustness because $(\eta_l)$ and $(\eta_u)$ are not jointly varied across datasets, imbalance ratios, and GNN backbones. The authors should provide a 2D sensitivity heatmap over $((\eta_l,\eta_u))$, report minority-class recall and pseudo-label precision in addition to F1/bAcc, and compare directly against standard high-confidence pseudo-labeling without an upper threshold. This would substantiate whether the upper threshold is genuinely beneficial rather than an arbitrary tuning choice.

2. The authors do state in the theory section and conclusion that the current theoretical analysis is binary. However, the connection between the binary theory and the multi-class empirical setting remains overstated. In particular, the discussion should avoid implying that Theorem 4.6 directly covers the multi-class experiments. The theory should be framed as binary/two-class motivation for minority pseudo-label augmentation, unless the authors extend the bound to a genuine multi-class setting.